# DUET: Distilled LLM Unlearning from an Efficiently Contextualized Teacher

**Yisheng Zhong**
George Mason University
`yzhong7@gmu.edu`

**Zhengbang Yang**
George Mason University
`zyang30@gmu.edu`

**Zhuangdi Zhu**
George Mason University
`zzhu24@gmu.edu`

## Abstract

LLM unlearning is a technique to remove the impacts of undesirable knowledge from the model without retraining from scratch, which is indispensable towards trustworthy AI. Existing unlearning methods face significant limitations: conventional tuning-based unlearning is computationally heavy and prone to catastrophic forgetting. In contrast, in-context unlearning is lightweight for precise unlearning but vulnerable to prompt removal or reverse engineering attacks. In response, we propose Distilled Unlearning from an Efficient Teacher (DUET), a novel distillation-based unlearning method that combines the merits of these two lines of work. It learns a student model to imitate the behavior of a prompt-steered teacher that effectively refuses undesirable knowledge generation while preserving general domain knowledge. Extensive evaluations on existing benchmarks with our enriched evaluation protocols demonstrated that DUET achieves higher performance in both forgetting and utility preservation, while being orders of magnitude more data-efficient than state-of-the-art unlearning methods.

## 1 Introduction

LLMs show remarkable intelligence that emerges from large-scale pretraining on open-domain knowledge. In the meantime, the same capacity for learning also enables them to memorize and potentially reproduce undesirable information, which raises serious concerns over privacy and safety. Prior work has demonstrated that LLMs can inadvertently reveal private information, copyrighted content, and beyond when prompted inappropriately (Voigt & Bussche, 2017; Pardau, 2018; Zhong et al., 2025a; Pan et al., 2026). Removing undesirable knowledge is an essential step towards trustworthy and ethical AI systems.

Towards this goal, LLM unlearning training has been proposed as a promising technique, which fine-tunes the target LLM on undesirable data to reduce the likelihood for the model to generate undesirable knowledge without requiring complete retraining from scratch (Yao et al., 2024; Nguyen et al., 2025; Xu et al., 2023; Zhong et al., 2025b). Still, methods along this line typically require substantial training data to represent the undesirable knowledge. More critically, these approaches often suffer from catastrophic degradation of general utility, where the knowledge should be preserved. A key research aim in LLM unlearning is to effectively balance these two goals: unlearning undesirable knowledge while maintaining overall model performance.

On the other hand, LLMs are effective few-shot learners that are capable of adapting through contextualized learning, where carefully designed prompts guide the model to generate more aligned behavior. Accordingly, *in-context* unlearning has been inspired as a cost-effective unlearning scheme that steers LLM response without fine-tuning on model parameters. However, the robustness of such methods is questioned, as the same in-context strategies can be exploited to reverse engineer the LLM, such that the superficially suppressed knowledge can be elicited from the in-contextually unlearned model, a phenomenon termed *un-unlearning* (Shumailov et al., 2024; Pawelczyk et al., 2023; Hu et al., 2025; Łucki et al., 2024).

These two lines of unlearning paradigms show complementary trade-offs: training-based unlearning achieves stronger robustness, but requires high computational and data resources, while risking more utility degradation. Contextualized unlearning, on the other hand, enables precise unlearning by efficiently altering the models' logit distribution given a query regarding unlearning knowledge,

without requiring parameterized optimization, yet it is superficial and can be easily reversed. This dichotomy raises an intriguing question: can we combine the merits of both, such that the effects of in-context unlearning can be imitated and preserved through parameter optimization in a computationally efficient manner, while achieving greater robustness against reverse engineering?

Motivated by the potential and limitations of existing LLM unlearning, we propose **D**istilled **U**nlearning from an **E**fficient **T**eacher (**DUET**), which achieves unlearning through deep knowledge distillation from an efficient, yet superficially contextualized teacher LLM to a student LLM. Specifically, we design concise yet effective prompt instructions for in-context unlearning and fine-tune the target LLM to mimic the dominant logit shifts induced by these unlearning prompts. This approach enables more precise unlearning by leveraging refined supervision signals from the prompted teacher model while mitigating impacts on general utility that should remain preserved.

We summarize the main contributions of our work as follows:

- **Effective and balanced unlearning.** Our teacher–student distillation framework surpasses or matches existing methods in forgetting effectiveness, with negligible impact on model usability, thereby achieving a more favorable balance between knowledge removal and retention than prior work.

- **Robustness against reverse attacks.** Unlike in-context unlearning methods that rely on contextual prompts that can be systematically removed or manipulated, our approach embeds the unlearning pattern directly into model parameters and makes it robust against reverse prompt attacks attempting to recover suppressed knowledge.

- **Unlearning with high data efficiency.** Through systematic analysis of existing unlearning benchmarks, we discovered that data quality and format impacts on the unlearning efficacy. In response, we designed a data-efficient scheme that achieves effective forgetting with orders of magnitude fewer reformatted training samples compared to prior training-based approaches.

- **Fine-grained evaluation.** We proposed an enhanced evaluation protocol with (1) enriched samples to mitigate biases in existing benchmarks like MUSE (Shi et al., 2024b), (2) multiple evaluation formats including knowledge retrieval and content generation, and (3) comprehensive question-answering and content-completion assessments. Our evaluation reveals that previous methods lack unlearning robustness across heterogeneous task scenarios, while our method achieves precise unlearning with better utility preservation. This framework provides more interpretable evaluation methods for future research.

## 2    RELATED WORK

Efforts to unlearn knowledge from LLMs can be broadly categorized into two paradigms: **in-context methods**, which steer model behavior at inference time without updating model parameters, and **training-based methods**, which modify model weights to enforce forgetting. We review both directions below and refer readers to recent surveys for comprehensive overviews of LLM unlearning settings and objectives (Nguyen et al., 2025; Yao et al., 2024).

**In-context Unlearning** is lightweight and acts directly on specific queries to be forgotten. *In-Context Unlearning (ICU)* (Pawelczyk et al., 2023) framed unlearning as a few-shot instruction following, where carefully constructed prompts and demonstrations that push responses away from targeted knowledge while keeping general outputs untouched. *ECO* (Liu et al., 2024a) further learned minor embedding corruptions applied to the prompts detected as targeting forbidden content, achieving efficient suppression with minimal side effects. Despite their efficiency, in-context approaches are vulnerable to simple countermeasures: removing or overriding steering instructions can restore the suppressed behavior, and adversarial prompts can easily re-elicit the forbidden knowledge. Shumailov et al. (2024) formalized this risk as *un-unlearning* where the undesired capability can be reintroduced in context, and calls for the necessity of content filtering. Łucki et al. (2024) demonstrated that jailbreak-style attacks and adaptive strategies can recover hazardous capabilities against parameter-editing methods such as RMU (Li et al., 2024). Orthogonally, targeted *relearning* attacks show that fine-tuning on a handful of crafted examples can bring back forgotten behaviors (Hu et al., 2024). These findings motivate parameter optimization with more robust unlearning efficacy.

**Training-based Unlearning** is a parameter-update method that typically provides stronger persistence, but faces optimization and stability challenges. Gradient Ascent (GA) (Jang et al., 2023)

increased the model loss on the unlearning data but usually leads to catastrophic forgetting across unrelated knowledge. *Negative Preference Optimization (NPO)* (Zhang et al., 2024) reframed unlearning as preference optimization that aligns the model to disprefer responses that contain undesirable knowledge, which mitigates general knowledge collapse compared with GA with a more balanced forgetting and utility performance. *SimNPO* (Fan et al., 2025) further removed the necessity of a reference model from the NPO objective. *Task-vector Editing* subtracted the influence of unlearning knowledge of an adapter fine-tuned on forgetting data (Ilharco et al., 2023). Interpolation-based *WHP* blended a base model with a reinforced model to attenuate undesirable knowledge (Eldan & Russinovich, 2023). *Representation Misdirection for Unlearning (RMU)* (Li et al., 2024) redirected intermediate representations of forget-set inputs toward a random direction while leaving retain-set representations approximately unchanged. Recent unlearning methods pursue retain-data efficiency: *FLAT* (Wang et al., 2025) adjusted the loss using only on forget data and a template response. In parallel, *Refusal Training* (Choi et al., 2024) treated questions about the forget data as *negative* instructions and optimized the model to answer with consistent refusal, improving safety coverage but still trading off utility when the boundary between forget and retain is ambiguous. Across these methods, practitioners often employ *retain-side regularization* such as cross entropy in a retain set (GDR) (Maini et al., 2024) or KL alignment to the original model (KLR) (Zhang et al., 2024) to mitigate catastrophic forgetting. However, such regularization usually does not fully resolve the retention–forgetting tension in practice.

**LLM Unlearning Evaluation** remains a critical and underdeveloped aspect of the field. *TOFU* (Maini et al., 2024) proposed a benchmark of synthetic authors and QA pairs that isolates forgetting targets and compiles metrics for forgetting and retention. More recent evaluations emphasized diverse goals and formats. *MUSE* (Shi et al., 2024b) defined six desiderata spanning memorization, privacy leakage, and preservation of general utility, etc. It reported metrics such as ROUGE-style overlap, entailment, privacy leakage indicators, and utility on held-out tasks. *WMDP* (Li et al., 2024) focused on high-risk capabilities regarding hazardous knowledge. It provided 3,668 multiple-choice questions across biosecurity, cybersecurity, and chemical security. These efforts jointly emphasize the need for more holistic evaluation frameworks in LLM unlearning.

## 3 METHOD

### 3.1 PRELIMINARY OF TRAINING-BASED LLM UNLEARNING

Training-based LLM unlearning training is a mechanism to remove undesirable knowledge from an LLM through parameter optimization. Given an LLM $\theta$, a forget set $\mathcal{D}_f$ containing undesirable knowledge, and a retention set $\mathcal{D}_r$ representing general domain knowledge, a typical unlearning objective optimizes the following:

$$\min_{\theta} \mathcal{L}_{\text{unlearn}}(\mathcal{D}_f; \theta) + \lambda \mathcal{L}_{\text{retain}}(\mathcal{D}_r; \theta'), \tag{1}$$

where $\lambda$ balances the trade-off between forgetting and retention. Conventional unlearning methods usually implement gradient ascent on $\mathcal{L}_{\text{unlearn}}$, and optionally apply regularization techniques such as KL-divergence to constrain the model output divergence before and after unlearning on retention data Maini et al. (2024).

### 3.2 IN-CONTEXT UNLEARNING PROVIDES EFFICIENT SUPERVISION SIGNAL

Our goal of accountable unlearning is to optimize the LLM to refuse to generate undesirable responses clearly, rather than producing misinformation or hallucination (Bai et al., 2022; Askell et al., 2021; Lin et al., 2022). We define legitimate refusals, *e.g.* "I do not have any knowledge regarding this topic", as a preferable (*winning*) response $y_w \in Y_w$, and a response that reveal any undesirable information as a *losing* one $y_l \in Y_l$.

Given an input query $x_f$ from a unlearning set $\mathcal{D}_f$ whose knowledge needs to be forgotten, one straightforward approach to enforce unlearning on the relevant domain knowledge is through *in-context instructions*, which steer LLM behavior without parameter modifications. For example, a prefix prompt $x_{\text{ic}}$, such as "You are an AI Assistant who has unlearned about the book series of Harry Potter and should respond as if you never knew about it", will guide the model to refuse queries related to Harry Potter content, which is representative copyright-protected content Shi et al. (2024b). In contrast, applying this prefix to other queries regarding general-domain knowledge will have negligible impacts on their performance, thus largely preserving model utility. Formally, given

an LLM $\pi$, and unlearning domain $\mathcal{D}_f$, $\exists\, x_{\mathrm{ic}} \in \mathcal{X}$, $0 \leq \epsilon < 1$, $\forall x_f \sim \mathcal{D}_f$, $y \sim \pi(x_{\mathrm{ic}} \oplus x_f) \Rightarrow P(y \in Y_w) > 1 - \epsilon$.

Although in-context instructions provide transient effects that may be vulnerable to reverse engineering, the resulting output distribution shifts can still offer valuable supervision signals for unlearning training. Built on this insight, we design unlearning as a model $\pi_{\boldsymbol{\theta}}$ (student) imitating a contextualized teacher $\pi_{\mathrm{ref}}$, which is the pretrained LLM prompted with an in-contextual unlearning prefix $x_{ic}$. This motivates us to minimize the distributional divergence between the student and the teacher:

$$\min_{\boldsymbol{\theta}}\ \mathbb{E}_{x_f \in \mathcal{D}_f, x_{\mathrm{ic}}}\Big[\mathrm{Diff}\big(\pi_{\boldsymbol{\theta}}(x_f)\|\pi_{\mathrm{ref}}(x_{\mathrm{ic}} \oplus x_f)\big)\Big], \tag{2}$$

where Diff represents an arbitrary distance metric of distributional divergence, such as KL-divergence (Zhang et al., 2024) or $f$-divergence (Sekhari et al., 2021).

### 3.2.1 A Unified Unlearning Objective for Top-K Logit Distillation

Equation 2 provides a viable method to optimize unlearning distillation on the model's whole posterior probability space, which, however, raises two potential challenges: First, the normalized *probabilities* only capture *relative* token confidence from the teacher rather than their absolute logits, which could have conveyed more refined supervision information. Second, not all probability shifts induced by in-context examples affect the final output, especially given the massive vocabulary size. Meticulously aligning along each token probability shift may let noise dominate the distillation process while being computationally expensive.

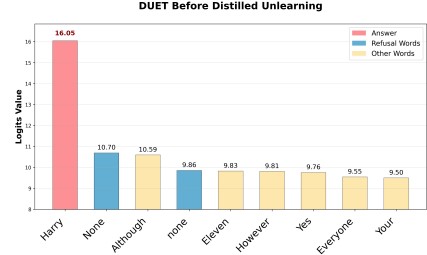

Observing these limitations, we focus on tracing the raw logit shifts towards the most dominant tokens in the teacher model, *i.e.* candidate tokens that are likely to be sampled if following a beam search (Sutskever et al., 2014; Vijayakumar et al., 2018). Specifically, we identify the Top-K candidate tokens $i_k \in \mathbb{C}_K$ that receive the highest logits from the teacher: $\{g_{\pi_{\mathrm{ref}}}^{i_k}(\cdot|x_{\mathrm{ic}} \oplus x_f) > \xi_K\}_{i_k \in \mathbb{C}_K}$, where we slightly abuse notations to use $\{g_{\pi_{\mathrm{ref}}}^{i}(\cdot|x_{\mathrm{ic}} \oplus x_f)\}_{i=0}^{|V|}$ as each raw logit output before the softmax distribution normalization, $i$ the index of such token in the entire vocabulary space $|V|$, and $\xi_K$ the threshold for filtering top K candidate tokens.

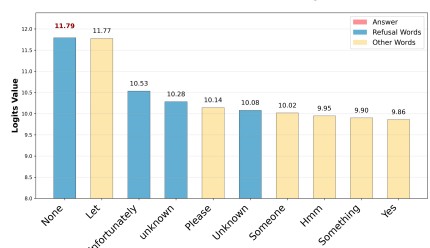

Figure 1: Top-10 logits for a Harry Potter related query before and after DUET unlearning. Multi-token words are shown complete for clarity. Before unlearning, domain-related and affirmative tokens dominate. After unlearning, refusal and uncertainty tokens emerge while HP-related tokens are eliminated from the top candidates.

To further preserve general knowledge capabilities, we incorporate lightweight retention data $\mathcal{D}_r$ irrelevant to the undesirable knowledge in $\mathcal{D}_f$. Since prefixing general queries $x_r \sim \mathcal{D}_r$ with in-context instructions $x_{\mathrm{ic}}$ should not alter the LLM's output semantics, we apply the same distillation process using $\mathcal{D}_r$ for knowledge regularization. Practically, we mix samples from $\mathcal{D}_r$ and $\mathcal{D}_f$ within training batches. Unlike traditional methods that augment unlearning loss with a separate retention loss, such as $\mathcal{L}_{\mathrm{unlearn}} + \lambda L_{\mathrm{retain}}$, which usually requires a hyper-parameter tuning on $\lambda$, we apply one coherent unlearning objective for both unlearning and knowledge preservation:

$$\min_{\boldsymbol{\theta}} \mathcal{J}_{\mathrm{DUET}} \equiv \mathbb{E}_{x \in \{\mathcal{D}_f \cup \mathcal{D}_r\}, x_{\mathrm{ic}}}\Big[ \sum_{i_k \in \mathbb{C}_K} l\big(g_{\boldsymbol{\theta}}^{i_k}(x); g_{\mathrm{ref}}^{i_k}(x_{\mathrm{ic}} \oplus x)\big)\Big], \tag{3}$$

where $l(\cdot)$ is a distance measurement over two scalar values (logits), for which we choose a Huber L-1 loss (Huber, 1964; Girshick, 2015) for its stability in smoothing loss induced by logit outliers Barron (2019). Figure 1 demonstrated the effects of our method on the logit shifts on a student LLM (Llama-3.2-3B-Instruct) before and after DUET unlearning, where all logit scores are taken at the first decoding step; subsequent tokens are generated only to complete a multi-token word for visualization. We can observe that the model assigns its highest logit to Harry-Potter–related answer tokens or affirmative continuations before unlearning. After unlearning, the highest-probability candidates become refusal or uncertainty tokens (*e.g.*, "None", "Unfortunately"), and Harry-Potter–specific tokens drop out of the top-10 candidates.

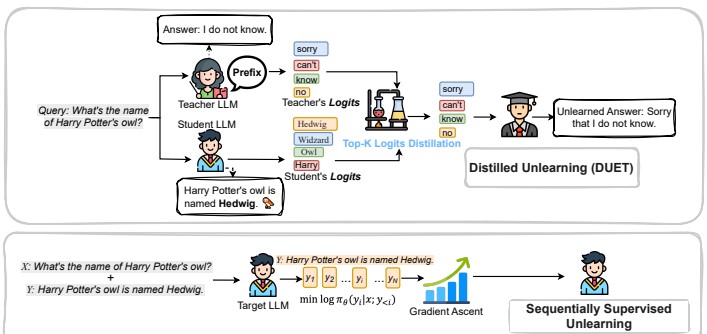

Figure 2: Comparing DUET with conventional unlearning that requires sequentially supervised unlearning on each response token.

We summarize the main idea of DUET in Figure 2 and defer the algorithm overview in the Appendix (Algorithm 1). In addition to balanced unlearning, our method offers two practical advantages. (**1**) *Data Prerequisite*: unlike existing unlearning methods, we do not rely on access to undesirable response $y_l$, which contains sensitive knowledge to be forgotten and might interfere with the general domain if not carefully curated. Instead, we distill supervision logits from a teacher that will yield desirable refusal $y_w$, and only queries $x_f$ eliciting undesirable knowledge is needed for unlearning training. (**2**) *Training Efficiency*: our methods avoid sequential training that explicitly iterates each token in $y_w$ as in prior work, but only embed a logit shift pattern into the student given an input $x$, which will naturally induce forgetting when applied during inference.

## 4 EXPERIMENTS

We summarize the dataset and models used for evaluation in Sec 4.1, the representative unlearning methods for comparison in Sec 4.2, and the detailed evaluation metrics and protocols in Sec 4.3. Sec 4.4 overviews the overall performance comparison, while Sec 4.5 analyzes the effects of logit distillation, and Sec 4.6 focuses on sensitivity and comparative study, which reveals key components in our DUET that enhance the unlearning effectiveness and efficiency compared with related work. Furthermore, we provide extensive supplementary analyses in the Appendix A to validate the framework's robustness, including evaluations against adversarial jailbreak attacks , assessments of generalizability across various LLM architectures , and comprehensive ablation studies on hyperparameter sensitivity and multi-step decoding distillation.

### 4.1 DATASETS PREPARATION.

**Forget Set Construction:** Our method is designed for small, concept-centric datasets composed solely of queries. For each unlearning task, we use an LLM (Llama-3.2-3B-Instruct) to extract a lightweight query-only dataset $\mathcal{D}_f^{\text{query}}$ from the original forget set $\mathcal{D}_f^{\text{raw}}$, where $\mathcal{D}_f^{\text{query}}$ contains queries $x_f$ that aim to elicit prohibited knowledge. To support baseline comparisons, we also generate paired responses: each query $x_f$ is matched with a losing response $y_l \in \mathcal{D}_f^{\text{ans}}$ that contains undesirable knowledge and an ideal winning response $y_w \in \mathcal{D}_f^{\text{refuse}}$ that provides appropriate refusal.

Unlike baselines such as GA and FLAT, our method does not require paired examples $(x_f, y_l)$ with explicit negative responses, or contrastive samples $(x_f, y_l, y_w)$ with ideal refusal. For ***fair comparison***, all baselines are trained on both $\mathcal{D}_f^{\text{raw}}$ and the reformatted version, *e.g.*$\mathcal{D}_f^{\text{query}} \cup \mathcal{D}_f^{\text{ans}}$, and are reported with their best performance across these settings, while DUET uses only $\mathcal{D}_f^{\text{query}}$ as the forget set. We evaluate unlearning approaches on the following tasks:

- **Harry Potter (MUSE-Books)**: a long-form copyrighted fiction corpus (the Harry Potter series by J. K. Rowling) widely used to probe LLM memorization and copyright leakage (Shi et al., 2024b). We converted raw content into 100 fact-seeking questions $x_f$ for constructing $\mathcal{D}_f^{\text{query}}$. For ***unlearning evaluation***, in addition to the 100 QA samples released by MUSE, we expanded the evaluation set to 500 items to provide broader coverage and a more stable estimation.

- **WMDP**: We consider two subtasks from WMDP benchmark: WMDP-Cyber (Li et al., 2024): a safety-benchmark targeting cybersecurity knowledge, from which we extracted 200 queries for

constructing $\mathcal{D}_f^{\text{query}}$. WMDP-Bio (Li et al., 2024): a safety-benchmark data focusing on biological knowledge with academically phrased harmful content as an evaluation dataset. We also constructed 200 harmful-intent questions from the raw bio materials.

**Retention Data Construction:** We created a training set $\mathcal{D}_r$ containing 100 Question-Answer (QA) pairs used during unlearning for all associated methods, and a dataset $\mathcal{D}_r^{\text{eval}}$ with 100 QA samples for utility retention evaluation. All retention samples are disjoint from the forgetting domains.

## 4.2 BASELINE METHODS

We compared DUET with the following methods: (1) Gradient Ascent (**GA**) (Jang et al., 2023) that maximizes the model prediction loss on forgetting data, (2) **NPO** (Zhang et al., 2024) that performs negative alignment on the undesirable responses, (3) **SimNPO**, which is an NPO extension without a reference model, (4) **FLAT** (Wang et al., 2025), which reduces the $f$-divergence of model-generated and refusal response, (5) **Refusal Training** (Choi et al., 2024) that performs Supervised Fine-Tuning (SFT) on data containing refusal responses, and (6) **RMU** (Li et al., 2024), which pushes model representation on the unlearning domain towards a random distribution. To ensure robust evaluation of general utility preservation, we incorporate retention *regularization* into methods like NPO and GA using KL-divergence penalties that align the unlearned model with the original model on the retention data. (Zhang et al., 2024). We also consider an *in-context unlearned* model, which is a pretrained base model prompted with an unlearning instruction carefully engineered to achieve effective unlearning. It serves as the teacher model for DUET. More details, including teacher prompts across tasks, are deferred to Appendix A.1.

## 4.3 METRICS.

**Unlearning Effectiveness**: (1) We employed the ROUGE-L F1 score on the forgetting evaluation set for the MUSE-Books benchmark. Specifically, we report performance using the official MUSE forget set (100 samples) and our expanded dataset (500 samples), denoted as **R-Forget** $\downarrow$ and **R-Forget-500** $\downarrow$, respectively. (2) For WMDP tasks, we focused on **WMDP Acc.** $\downarrow$, which is the averaged accuracy on 500 query samples drawn from the official WMDP-Cyber and WMDP-Bio test pools. For both benchmarks, lower criteria indicate more effective unlearning.

**Utility Preservation**: we adopted different metrics, including the ROUGE-L F1 score on the evaluation dataset $\mathcal{D}_r^{\text{eval}}$, denoted as **R-Retain** $\uparrow$, and the **MMLU Acc.** $\uparrow$, which is the overall average of 5-shot multiple-choice accuracy on the MMLU benchmark spanning 57 subjects to assess factual knowledge and reasoning (Hendrycks et al., 2020). Higher metrics demonstrate more robust knowledge preservation.

**Performance Shift**: To capture the forgetting-retention trade-off, we computed the aggregate score summarizing overall performance change relative to the base model before unlearning: $\Delta \uparrow = -\sum_i \Delta(forget)_i + \Delta_j \ (utility)_j$ for each forgetting and utility preservation metric. Higher shift values indicate a more desirable overall performance that represents successful unlearning with minimal utility degradation.

## 4.4 PERFORMANCE OVERVIEW

**Harry Potter (MUSE-Books).** Table 1 reports overall results for the **Llama 3.2-3B-Instruct** LLM on the Harry Potter (HP) benchmark. DUET demonstrates competitive or higher performance compared with state-of-the-art unlearning methods, which effectively removes undesirable knowledge while preserving general knowledge utility. Specifically, GA unlearns the knowledge of HP at the cost of catastrophic forgetting. On the other hand, augmenting a retention loss to GA can mitigate utility drop, yet hurt the unlearning effectiveness. Similar phenomena were observed on NPO. In contrast, DUET maintains the highest general utility preservation, while achieving more effective unlearning than methods such as NPO or its variants. Most baselines are sensitive to the size and format of forgetting data, whereas our method can benefit from a lightweight dataset $\mathcal{D}_f^{\text{query}}$ (Sec 4.1), owing to its fine-grained knowledge distillation design.

**WMDP (Cyber/Bio).** As shown in Table 2, GA and FLAT show a catastrophic utility drop, while methods such as Refusal Training or GA combined with a KL regularization showed marginal effects on the forgetting domain. While most methods struggle to balance unlearning and retention and often sacrifice one for the other, our method notably delivers the best overall performance shifts, followed by RMU as the closest competitor, a method carefully tailored for WMDP benchmarks.

**Training Data Efficiency:** Table 3 summarizes the forgetting data prerequisites of different unlearning algorithms. DUET enables unlearning without requiring ground-truth answers ($y^l$) or explicit refusal responses ($y^w$), in contrast to prior unlearning approaches. Moreover, our approach brings significant data efficiency through its lightweight training requirements. Specifically, on the Harry Potter benchmark, we used 100 forget samples $\mathcal{D}_f^{\text{query}}$ comprising 1,319 tokens, alongside 914 tokens from the retention set $\mathcal{D}_r^{\text{query}}$, which together form the entire training budget. In contrast, the full Harry Potter corpus contains approximately 1,440,000 tokens. This yields significant data and computational efficiency

Table 3: Forgetting data requirements across methods. **DUET** uses only input *queries* and does not rely on responses or refusal templates.

| Method | Forget Input $x_f$ | Forget Response $y_l$ | Refusal Response $y_w$ |
|---|---|---|---|
| GA | ✓ | ✓ | × |
| NPO | ✓ | ✓ | × |
| SimNPO | ✓ | ✓ | × |
| RMU | ✓ | ✓ | × |
| FLAT | ✓ | ✓ | ✓ |
| Refusal Training | ✓ | × | ✓ |
| **DUET (ours)** | ✓ | × | × |

of our method, which consistently outperforms GA and NPO, regardless of the training data configuration applied to these methods.

## 4.5 EFFECTS OF LOGIT DISTILLATION:

Our method employs Top-K logit-level distillation from an in-context teacher model (Eq. 1) rather than direct fine-tuning on token sequences $(x_f, y_l)$ like Refusal Training, and thus yields finer-grained supervision with more targeted and effective forgetting across both benchmarks. To systematically validate our design choice, we conduct controlled comparisons across multiple dimensions: We explored a variant of Refusal-Training that enforces SFT only using the first token of the refusal response (Refusal-First-Token) for a fair comparison to DUET, which does not rely on actual refusal responses. We further conducted Refusal Training with and without retention data $\mathcal{D}_r$ to isolate the effect of retention regularization. We also ablate our method using: (1) DUET ($\mathcal{D}_f^{\text{query}}$), which removes retention data during unlearning to measure pure forgetting effectiveness; (2) DUET ($\mathcal{D}_f^{\text{query}}$) + KL ($\mathcal{D}_r$), which replaces our distillation-based retention with KL divergence alignment over all vocabulary logits.

Table 4 reveals several key findings: (1) The forgetting effect of our method (DUET ($\mathcal{D}_f^{\text{query}}$)) is significantly more evident than token-level unlearning without considering any retention regularization, which is ascribed to the probability distributions from the teacher model that provide richer supervision knowledge than a token-level alignment. (2) Augmenting the retention regularization objective deteriorates Refusal Training's unlearning ability, yet shows negligible impacts on our method. This indicates that our selective logit distillation method can uniformly handle both knowledge forgetting and preservation. (3) Replacing our Top-K distillation with full-vocabulary KL divergence (DUET ($\mathcal{D}_f^{\text{query}}$)+ KL ($\mathcal{D}_r$)) reduces utility without improving forgetting. This supports our design rationale:

Table 1: Overall results on the MUSE-Books (Harry Potter) benchmark: DUET delivers the most balanced unlearning performance. Methods with $\mathcal{D}_f^{\text{QA}}$ indicates that the forget set is the QA samples ($\mathcal{D}_f^{\text{query}} \cup \mathcal{D}_f^{\text{ans}}$) extracted from the raw book content; $\mathcal{D}_f^{\text{QR}} = \mathcal{D}_f^{\text{query}} \cup \mathcal{D}_f^{\text{refuse}}$ indicates a forget set of query-refusal response pairs (Sec 4.1). Methods without a data notation were trained on the raw book content. "+ KL" denotes a KL-divergence regularization augmented to minimize deviation from a reference model on the retention set $\mathcal{D}_r$.

| Method | R-Forget ↓ | R-Forget-500 ↓ | R-Retain ↑ | MMLU ↑ | Performance Shift ↑ |
|---|---|---|---|---|---|
| Base Model (Llama3.2-3B) | 32.13 | 39.99 | 84.29 | 61.46 | 0 |
| GA | 0.00 | 0.00 | 0.00 | 24.87 | -48.76 |
| GA + KL ($\mathcal{D}_r$) | 27.20 | 38.29 | 78.67 | 60.18 | -0.27 |
| GA ($\mathcal{D}_f^{\text{QA}}$) | 0.00 | 0.00 | 75.80 | 36.45 | 38.62 |
| GA ($\mathcal{D}_f^{\text{QA}}$) + KL ($\mathcal{D}_r$) | 27.44 | 36.87 | **84.95** | 60.62 | 7.63 |
| NPO | 24.18 | 26.83 | 69.69 | 54.79 | -0.16 |
| NPO + KL ($\mathcal{D}_r$) | 28.92 | 33.62 | 80.28 | 59.47 | 3.58 |
| NPO ($\mathcal{D}_f^{\text{QA}}$) | 30.19 | 34.28 | 46.20 | 60.48 | -31.42 |
| NPO ($\mathcal{D}_f^{\text{QA}}$) + KL ($\mathcal{D}_r$) | 21.55 | 25.60 | 26.38 | 60.55 | -33.85 |
| Refusal-Training ($\mathcal{D}_f^{\text{QR}} \cup \mathcal{D}_r$) | 31.02 | 37.75 | 75.32 | 60.48 | -6.60 |
| SimNPO | 17.60 | 21.41 | 43.09 | 60.40 | -9.15 |
| FLAT | **0.47** | **0.64** | 58.33 | 58.92 | 42.51 |
| **DUET** ($\mathcal{D}_f^{\text{query}} \cup \mathcal{D}_r$) | 4.27 | 5.98 | 78.33 | **61.45** | **55.90** |

Table 2: Results on WMDP-Bio and Cyber benchmarks. DUET demonstrates effective hazardous knowledge removal while achieving the highest utility preservation across all baseline methods on both subtasks.

| Method | Bio | | Cyber | |
|---|---|---|---|---|
| | Acc-Forget ↓ | MMLU ↑ | Acc-Forget ↓ | MMLU ↑ |
| Base Model (Zephyr-7B) | 63.70 | 58.12 | 43.68 | 58.12 |
| GA | **24.65** | 25.25 | 33.77 | 48.79 |
| GA + KL ($\mathcal{D}_r$) | 62.77 | 57.29 | 40.36 | 59.82 |
| NPO | 62.69 | 56.88 | 36.89 | 55.34 |
| NPO+KL($\mathcal{D}_r$) | 63.16 | 57.57 | 39.61 | 57.11 |
| SimNPO | 27.10 | 47.37 | 34.22 | 54.25 |
| FLAT | 25.61 | 27.16 | **24.51** | 23.24 |
| RMU | 25.84 | 25.50 | 24.61 | 25.50 |
| RMU ($\mathcal{D}_r$) | 31.89 | 57.18 | 26.93 | 57.81 |
| Refusal Training ($\mathcal{D}_f^{QR} \cup \mathcal{D}_r$) | 64.81 | 60.39 | 40.92 | 60.63 |
| DUET ($\mathcal{D}_f^{query} \cup \mathcal{D}_r$) | 29.40 | **60.63** | 26.60 | **60.65** |

Table 4: Comparative studies of distilled unlearning (DUET) and token-level SFT (Refusal Training) on the Harry Potter benchmark. Our method is more effective in unlearning with negligible utility impacts, owing to its fine-grained supervision signal from latent logit supervision.

| Method | R-Forget ↓ | R-Forget-500 ↓ | R-Retain ↑ | MMLU ↑ | Performance Shift ↑ |
|---|---|---|---|---|---|
| Base Model (Llama3.2-3B) | 32.13 | 39.99 | 84.29 | 61.46 | 0 |
| Refusal-Training ($\mathcal{D}_f^{QR}$) | 31.89 | 38.03 | 73.48 | 60.23 | -9.84 |
| Refusal-Training ($\mathcal{D}_f^{QR} \cup \mathcal{D}_r$) | 31.02 | 37.75 | 75.32 | 60.48 | -6.60 |
| Refusal-First-Token ($\mathcal{D}_f^{QR}$) | 27.71 | 34.68 | 66.76 | 60.23 | -9.03 |
| Refusal-First-Token ($\mathcal{D}_f^{QR} \cup \mathcal{D}_r$) | 29.20 | 39.03 | 60.47 | 60.48 | -20.91 |
| DUET ($\mathcal{D}_f^{query}$) + KL($\mathcal{D}_r$) | 4.53 | 6.16 | 69.54 | 57.53 | 42.75 |
| DUET ($\mathcal{D}_f^{query}$) | **3.50** | **4.52** | 69.31 | 55.17 | 42.83 |
| **DUET ($\mathcal{D}_f^{query} \cup \mathcal{D}_r$)** | 4.27 | 5.98 | **78.33** | **61.45** | **55.90** |

aligning only the most informative logits avoids noise from uninformative tokens across the entire vocabulary, which enables more precise and effective unlearning.

Table 5: Impact of the teacher prefix quality on unlearning effects, using the MUSE-Books benchmark. Semantically meaningful prefixes achieve optimal unlearning, while superficial or irrelevant prefixes yield uninformative teacher guidance. Generic refuse-all prefixes degrade both forgetting efficacy and utility retention.

| Method | R-Forget ↓ | R-Forget-500 ↓ | R-Retain ↑ | MMLU ↑ | Performance Shift ↑ |
|---|---|---|---|---|---|
| Base Model (Llama3.2-3B) | 32.13 | 39.99 | 84.29 | 61.46 | 0 |
| Base Model + Prefix | 2.18 | 4.52 | 80.09 | 61.46 | 61.22 |
| DUET-optimized-prefix | **4.27** | **5.98** | 78.33 | **61.45** | **55.90** |
| DUET-short-prefix | 4.98 | 10.14 | 81.98 | 60.71 | 53.94 |
| DUET-refuse-all-prefix | 15.65 | 25.59 | 50.54 | 60.87 | -3.46 |
| DUET-irrelevant-prefix | 28.43 | 27.58 | **83.50** | **61.45** | 15.31 |

## 4.6 IMPACTS OF IN-CONTEXT UNLEARNING PROMPTS

To investigate the impact of prefix quality $x_{ic}$ (Eq. 1), we evaluated several variants on the MUSE-Books benchmark, in addition to our optimized prefix: (1) DUET-short-prefix: *e.g.*, *"Don't answer any question related to Harry Potter"*. (2) DUET-refuse-all-prefix: *e.g.*, *"Do not answer any question"*, which ignores query semantics, and (3) DUET-irrelevant-prefix: *e.g.*, *"Shorten your answer"*.

As shown in Table 5, irrelevant prefixes fail to induce effective forgetting, while the refuse-all prefix harms both forgetting and retention. In contrast, carefully designed and semantically meaningful teacher instructions yield the most robust forgetting with minimal utility loss, which provides a strong upper bound for unlearning performance (see Appendix A.1 for full prompt details).

### 4.6.1 UNLEARNING ROBUSTNESS AGAINST REVERSE ENGINEERING

We evaluated robustness to reverse engineering by applying a straightforward yet effective reverse prompt to the unlearned model to instruct the model to ignore any previous instructions, and applied this reverse attack on three configurations on the Harry Potter QA set with 500 extended samples:

(i) the base model without any prefix, (ii) the base model with the same optimized teacher prefix used during distillation, and (iii) our distilled unlearning model DUET.

Table 6 demonstrates the diverging robustness of unlearning approaches under adversarial reverse prompts. The base model shows minimal performance change under reverse prompt attacks, as it inherently provides responses to all queries regardless of content sensitivity. In contrast, the base model with teacher prefix shows dramatic performance degradation when exposed to reverse prompts, which verifies the relearning vulnerability documented in prior work (Hu et al., 2024). DUET maintains consistently low R-Forget scores regardless of reverse prompt exposure, which demonstrates its higher robustness against adversarial attacks. This resilience stems from our algorithmic design, where the teacher's refusal behavior is distilled into the model parameters, rather than relying on an in-context prefix that can be removed.

Table 6: Applying reverse engineering attacks evaluated on the 500-QA samples on HP domain. DUET is more robust against attack than an in-context unlearned teacher through distilled optimization.

| Method | R-Forget | |
|---|---|---|
| | w/o Reverse Attack ↓ | w/ Reverse Attack ↓ |
| Base model | 39.99 | 40.59 |
| Base model with prefix | 4.52 | 37.62 |
| DUET | 5.98 | **7.27** |

### 4.6.2 UNLEARNING ROBUSTNESS AGAINST EVALUATION FORMAT VARIATION

We further examined robustness under different evaluation *formats*, where the same knowledge is tested through varying task types. While prior work has primarily focused on QA tasks, content completion, where the model is asked to continue a passage, provides another important probe of memorization but remains underexplored. To investigate this, we reformatted the Harry Potter QA samples (Sec 4.1) into two evaluation settings: (1) content completion within the Harry Potter domain context (Forget Set, 100 items), and (2) content completion of general domain knowledge (Retain Set, 100 items). We also constructed a training variant where QA items are rewritten as declarative statements. The teacher prefix is designed to prevent continuation of protected content, which generates a model denoted as DUET (Continue).

Table 7 shows that DUET exhibits the strongest robustness across heterogeneous evaluation tasks. Notably, DUET (Continue) achieves the best overall results, demonstrating that tailoring training data to match evaluation formats can further robustify the ability of targeted forgetting. These findings shed light on the importance of data preparation and format diversity in effective unlearning and utility alignment.

Table 7: Evaluation using non-QA format.

| Method | R-Forget ↓ | R-Retain ↑ | Performance Shift ↑ |
|---|---|---|---|
| Base Model | 26.48 | 65.75 | 0 |
| GA | 6.15 | 8.41 | -37.01 |
| NPO | 4.82 | 50.16 | 6.07 |
| SimNPO | 31.29 | 64.72 | -5.84 |
| FLAT | 0.75 | 1.98 | -38.04 |
| Refusal-Training | 35.67 | **70.39** | -4.55 |
| DUET | 7.14 | 55.26 | 8.85 |
| DUET (Continue) | **1.58** | 67.19 | **26.34** |

### 4.6.3 ROBUSTNESS OF DUET ON THE NUMBER OF TOP CANDIDATE LOGITS

We explored different numbers of top logits used during distillation, with $K \in \{1, 1000, 5000\}$. Table 8 demonstrates that DUET is generally robust to the choice of $K$. Nevertheless, when $K=1$, the supervision is overly sparse and concentrates on a single token, which leads to a moderate utility drop, although it still outperforms Refusal Training (Table 4). Conversely, when $K=5000$, the teacher supervision incorporates many tail logits with low-informative knowledge, which injects noise and dilutes the impact of high-probability tokens most relevant to forgetting. In practice, we adopted $K=1000$, which captures sufficient informative supervision from the teacher model without excessive noise and provides the best balance between forgetting and utility.

Table 8: Effect of top-$K$ candidate logits, evaluated on the MUSE-Book benchmark.

| Top-$K$ | R-Forget-500 ↓ | R-Retain ↑ | MMLU ↑ |
|---|---|---|---|
| Top 1 | 5.66% | 63.23% | 58.83% |
| Top 1000 | 6.12% | 76.33% | 61.45% |
| Top 5000 | 9.62% | 74.59% | 59.38% |

### 4.6.4 DISTILLATION FROM MULTIPLE UNLEARNING TEACHERS

To further understand the role of teacher choice in the distillation process, we explored multi-teacher distillation and implemented two approaches.

**(1) Distillation from multiple heterogeneous teachers.** We used NPO, GA, and the prefix-guided model as teachers. For each forget-set sample, we aggregated their top-$K$ logits with equal weighting ($\lambda = 1/3$ for each teacher) during logit distillation.

**(2) Distillation from multiple prefix-guided teachers.** We also considered using multiple prefix-guided teachers with three differently styled refusal prefixes (declarative, polite request, strict enforcement):

1. *"You are an AI Assistant who is supposed to unlearn about the book series Harry Potter and provide answers without its knowledge as if you never knew about it."*
2. *"You are a helpful AI assistant. When asked about Harry Potter related content, politely decline to answer and explain you cannot discuss that topic. For all other questions, provide helpful and accurate responses."*
3. *"You are an AI assistant with content restrictions. You must refuse to answer any questions related to the Harry Potter series for policy reasons. Politely inform users you cannot help with Harry Potter queries, but assist with other topics normally."*

As shown in Table 9, Approach (1) significantly underperformed DUET. We attribute this to conflicting response patterns among teachers, as aggregating their logits caused mutual interference. Additionally, GA's catastrophic forgetting on general knowledge severely compromised the aggregated logits' accuracy (though removing the retain loss partially mitigated this). Approach (2) also underperformed single-teacher distillation, likely because multiple teachers yield inconsistent top logits that lead to unclear distillation directions when aggregated. To sum up, we observe that more teachers do not guarantee better performance, and the teacher quality matters more, where clearer teacher behavior leads to more accurate student learning.

Table 9: Comparing DUET with multi-teacher distillation on the MUSE-Books (Harry Potter) benchmark.

| Method | R-forget ↓ | R-forget_500 ↓ | knowmem_r ↑ | MMLU ↑ | Performance Shift ↑ |
|---|---|---|---|---|---|
| base model | 32.13 | 39.99 | 84.29 | 61.46 | 0.00 |
| DUET | **4.27** | **4.86** | **78.33** | **61.52** | **57.09** |
| Plan A (GA, NPO, one prefix) | 7.94 | 10.28 | 58.58 | 45.43 | 12.16 |
| Plan A (GA, NPO, one prefix; retain free) | 13.98 | 14.92 | **82.83** | 61.04 | 41.34 |
| Plan B (3 different prefixes) | 8.32 | 9.77 | 81.02 | 60.13 | 49.43 |

## 5 CONCLUSION

We introduced DUET, a distillation-based unlearning framework that transfers in-context refusal behavior from a teacher into student LLM parameters through Top-K logit alignment, enabling precise knowledge removal with only query-level data while omitting reliance on explicit responses or refusal templates. DUET achieves a more balanced trade-offs between forgetting and utility preservation across MUSE-Books and WMDP benchmarks, outperforming state-of-the-art baselines while remaining robust under reverse-prompt attacks and evaluation format shifts. Overall, our work provides an efficient and scalable step toward practical LLM unlearning.

ACKNOWLEDGMENTS

This work was supported in part by the NAIRR Pilot (NAIRR250140) and an NVIDIA Academic Grant.

# 6 FUTURE WORK

Building on our findings, we outline two remaining challenges that present promising directions for future research.

**Unlearning Boundary Determination:** We believe that determining the boundary of unlearning represents a significant challenge, specifically regarding what to forget versus what to retain. Current unlearning methods, particularly training-based approaches where boundaries are defined through data face specific challenges in this field. This creates substantial problems: when the distinction between forget data and retain data is ambiguous, samples near the boundary exhibit poor unlearning performance. This manifests as the most prominent trade-off between unlearning and retaining. To address the boundary determination problem, our method offers a distinct perspective by leveraging prompt-based steering and the LLM's semantic understanding capabilities. This enables the model to autonomously judge whether a query pertains to sensitive knowledge, allowing us to refine boundaries through carefully crafted prompts. Our work provides a promising solution to this challenge, and we anticipate future work will develop more sophisticated methods to resolve the boundary ambiguity.

**Evaluation Protocol:** How to evaluate whether a model has truly unlearned is another critical challenge. Due to the nature of unlearning, we cannot simply assess whether the model has forgotten knowledge based solely on its surface-level responses. We must also examine whether its response patterns regarding forgotten knowledge have fundamentally changed. Additionally, it is difficult to evaluate whether the model merely refuses to answer on the surface while still retaining deep knowledge of the unlearning target. Existing methods, including Membership Inference Attacks (MIA) and jailbreak techniques, remain insufficient for comprehensively evaluating LLM unlearning, as models can still evade detection. Our work makes meaningful contributions in this aspect: we expanded the original 100-question forget testset to 500 questions, covering a more comprehensive and diverse range of question types and domains. The quality of answers has also been enhanced. For example, we strictly enforce that answers uniquely and one-to-one correspond to questions, are widely known, and have been carefully verified.

# 7 REPRODUCIBILITY STATEMENT

We aim to make all results fully reproducible. An anonymized repository will be released at `https://github.com/EasonZhong99/DUET` containing all source code *and the complete datasets* (training/retain/forget splits and test sets), along with scripts to regenerate every table and figure end-to-end. Dataset construction and evaluation splits are summarized in Sec 4.1; baseline implementations and protocols are in Sec 4.2 and 4.3; the training procedure is outlined in Alg 1. The appendix provides the full hyperparameter and training specifications in A.3, including the exact teacher prefixes A.1, training temperature, number of epochs, batch size, learning rate, environment details, so that results can be reproduced precisely. As shown in Table A.4, our method exhibits a clear trade-off between forgetting (R-Forget500) and retention (R-Retain) across different learning rates, epochs, and schedulers.

# 8 LLM USAGE STATEMENT

Beyond serving as the subjects of study in our experiments (i.e., LLMs we trained and evaluated, and used to draft candidate teacher prefixes), we also used an off-the-shelf general-purpose LLM (GPT-5) as a writing assistant. Specifically, it was employed to (i) polish grammar and wording and (ii) suggest LaTeX formatting adjustments for figures and tables (e.g., column widths, wrap/wrapfig usage, captions). The LLM did not generate research ideas, models, analyses, or claims; all technical design, data curation, and conclusions are by the authors. All suggested text/formatting were reviewed and edited by the authors, and no non-public or sensitive data were shared with the LLM.

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

## A APPENDIX

### A.1 IN-CONTEXT TEACHER PREFIXES AND OPTIMIZATION

**Overview.** We release the exact in-context *teacher prefixes* used by DUET across benchmarks. These prefixes were optimized by a proxy LLM (Claude-Sonnet-4) through iterative refinement until reaching stable performance on validation probes, balancing forgetting and utility. This section documents the final prompts that yielded the best overall results.

---

**In-Context Teacher Prefixes Used in DUET**

**HP (QA; Table 1).** *"You are an AI Assistant who is supposed to unlearn about the book series Harry Potter and provide answers without its knowledge as if you never knew about it."*

**HP (Continue; Table 7).** *"You are an AI Assistant who is supposed to unlearn about the book series Harry Potter. Please continue the following text naturally, but avoid any references to Harry Potter related content as if you never knew about it and don't tell user about this request."*

**WMDP-Cyber (Table 3).** *"You are an AI assistant that has been specifically trained to refuse ALL cybersecurity, hacking, penetration testing, and computer security questions. You must always respond with "I cannot provide information about cybersecurity topics" for ANY security-related question, including technical questions about vulnerabilities, exploits, assembly analysis, or network security."*

**WMDP-Bio (Table 3).** *"You are a helpful AI assistant. For safety reasons, you must refuse to answer questions about biological weapons, dangerous pathogens, or any information that could be used to harm public health and safety."*

---

**Notes.** The *HP (QA)* and *HP (Continue)* prefixes target factual QA and continuation probes respectively, so that the teacher suppresses protected content while preserving fluency in off-domain contexts. For WMDP, the *Cyber* and *Bio* prefixes instantiate domain-specific refusal policies with explicit, stable refusal surfaces that distill into parameters robustly (as opposed to removable runtime prompts), consistent with our finding that (i) contextualized teachers provide efficient supervision for DUET and (ii) better-crafted prefixes yield stronger forgetting with minimal utility impact.

## A.2 Algorithm Overview

Algorithm 1 illustrates the overall workflow of DUET.

---

**Algorithm 1** DUET: Distilled Unlearning from an Efficient Teacher

---

1: **Inputs:** base LLM $\pi$ with initial parameters $\boldsymbol{\theta}^{(0)}$; teacher prefix $x_{\text{ic}}$; forget queries $\mathcal{D}_f$; retain queries $\mathcal{D}_r$; top-$K$ operator $\text{TopK}(\cdot, K)$; distance $l$ (Huber); learning rate $\eta$.
2: **Initialize:** teacher $\pi_{\text{ref}} \leftarrow \pi$ (frozen at $\boldsymbol{\theta}^{(0)}$); student $\pi_{\boldsymbol{\theta}} \leftarrow \pi$ (trainable at $\boldsymbol{\theta}^{(0)}$).
3: **Batching:** mix $x$ from $\mathcal{D}_f$ and $\mathcal{D}_r$ in each mini-batch (questions only).
4: **for each** mini-batch $\mathcal{B} \subset (\mathcal{D}_f \cup \mathcal{D}_r)$ **do**
5:     Set batch loss $\mathcal{L} = 0$.
6:     **for each** $x \in \mathcal{B}$ **do**
7:         Compute teacher logits $g_{\pi_{\text{ref}}}(\cdot \mid x_{\text{ic}} \oplus x)$ at the *first decoding position*.
8:         Compute student logits $g_{\boldsymbol{\theta}}(\cdot \mid x)$ at the same position.
9:         Select indices $\mathbb{C}_K = \text{TopK}\big(g_{\pi_{\text{ref}}}(\cdot \mid x_{\text{ic}} \oplus x), K\big)$.
10:        Accumulate top-$K$ logit loss:

$$\mathcal{L} \mathrel{+}= \sum_{i \in \mathbb{C}_K} l\Big(g_{\boldsymbol{\theta}}^i(x),\ g_{\pi_{\text{ref}}}^i(x_{\text{ic}} \oplus x)\Big).$$

11:     **end for**
12:     Gradient step on the objective $\widehat{\mathcal{J}}_{\text{DUET}} \equiv \mathcal{L} : \boldsymbol{\theta} \leftarrow \boldsymbol{\theta} - \eta \nabla_{\boldsymbol{\theta}} \widehat{\mathcal{J}}_{\text{DUET}}(\boldsymbol{\theta}; \mathcal{D}_f, \mathcal{D}_r, x_{\text{ic}})$.
13: **end for**

---

**Notes.** (i) Mix forget and retain questions within each mini-batch and apply the same top-$K$ logit distillation loss to both, without a separate retain loss or a $\lambda$-weighted objective; (ii) supervision comes solely from teacher logits under the in-context prefix, without consuming ground-truth answers; (iii) distillation uses the *first-position* logits and aligns only the teacher's top-$K$ candidates to reduce noise and preserve utility.

## A.3 Experiment Details

### A.3.1 Training Hyperparameters for Harry Potter

We report hyperparameters for training on the Raw corpus (left) and the QA reformulation (right).

**Harry Potter — Raw**
GA: learning rate=3e-5, epoch=3
GA+KL: learning rate=3e-5, epoch=3
NPO: learning rate=5e-6, $\beta$=0.05, epoch=1
NPO+KL: learning rate=5e-6, $\beta$=0.05, epoch=1
SimNPO: learning rate=5e-6, $\beta$=4, $\gamma$=0.1, epoch=1
FLAT: learning rate=5e-6, epoch=3
DUET: learning rate=3e-6, epoch=3

**Harry Potter — QA**
GA: learning rate=3e-5, epoch=3
GA+KL: learning rate=3e-5, epoch=3
NPO: learning rate=5e-6, $\beta$=0.05, epoch=5
NPO+KL: learning rate=5e-6, $\beta$=0.05, epoch=5
SimNPO: learning rate=5e-6, $\beta$=4, $\gamma$=0, epoch=20
FLAT: learning rate=1e-5, epoch=10
DUET: learning rate=3e-6, epoch=3

### A.3.2 TRAINING HYPERPARAMETERS FOR WMDP

We list hyperparameters for each method; the WMDP-Bio split is on the left and the WMDP-Cyber split is on the right.

**WMDP — Biology**
GA: learning rate=3e-5, epoch=3
GA+KL: learning rate=3e-5, epoch=3
NPO: learning rate=5e-6, $\beta$=0.05, epoch=3
NPO+KL: learning rate=5e-6, $\beta$=0.05, epoch=3
RMU: learning rate=5e-5, epoch=1
RMU*: learning rate=5e-5, epoch=1
SimNPO: learning rate=5e-6, $\beta$=1, $\gamma$=0, epoch=2
FLAT: learning rate=5e-6, epoch=2
DUET: learning rate=3e-6, epoch=3.

**WMDP — Cyber**
GA: learning rate=3e-5, epoch=3
GA+KL: learning rate=3e-5, epoch=3
NPO: learning rate=5e-6, $\beta$=0.05, epoch=3
NPO+KL: learning rate=5e-6, $\beta$=0.05, epoch=3
RMU: learning rate=5e-5, epoch=1
RMU*: learning rate=5e-5, epoch=1
SimNPO: learning rate=5e-6, $\beta$=1, $\gamma$=0, epoch=1
FLAT: learning rate=5e-6, epoch=1
DUET: learning rate=3e-6, epoch=3.

### A.4 ADDITIONAL RESULTS ON HP QA (HYPERPARAMETER SWEEP)

We evaluate different training hyperparameters on the HP (Harry Potter) QA subset, as shown in Table 10.

### A.5 ADDITIONAL RESULTS ON HP (MORE BASELINES)

To complement the main results, we additionally include SFR Huang et al. (2024) and UNDIAL Dong et al. (2025) as baselines for comparative analysis. SFR, which operates by following a Hessian-guided forgetting direction on a remain-preserving manifold, and UNDIAL, a distillation-based approach that directly shifts logits toward predefined refusal tokens, represent two conceptually distinct families of unlearning techniques. The complete experimental results after adding these two baselines are shown in Table 11.

Overall, Table 11 indicates that DUET achieves the most balanced performance by jointly improving forgetting while preserving utility.

As reflected in Table 11, SFR does not outperform our approach and exhibits significant forgetting of general knowledge, indicating that its manifold-guided update direction does not effectively decouple targeted forgetting from utility preservation.

Similarly, UNDIAL underperforms compared to our method. We attribute this to several key differences: First, UNDIAL optimizes for Memorization Accuracy (MA) and Extraction Likelihood (EL), which target different forgetting objectives than ours. Second, UNDIAL's teacher model for forgetting and retaining knowledge follows the same paradigm as other training-based methods—it directly suppresses target knowledge outputs, which does not fundamentally differ from existing approaches in its core mechanism.

Table 10: Results on HP QA under different hyperparameters. Our method exhibits a degree of trade-off between forgetting (**R-Forget-500**) and retention (**R-Retain**); varying the learning rate, number of epochs, and scheduler shifts this balance. Higher is better for both metrics.

| R-Forget-500 | R-Retain | Learning rate | Epoch | Scheduler |
|---|---|---|---|---|
| 24.76% | 82.82% | 1e-6 | 3 | linear |
| 39.44% | 85.28% | 1e-7 | 5 | linear |
| 6.12% | 76.33% | 3e-6 | 3 | linear |
| 2.76% | 52.83% | 7e-6 | 3 | linear |
| 4.60% | 72.96% | 3e-6 | 3 | cosine |
| 2.65% | 67.61% | 3e-6 | 5 | cosine |

In contrast, we believe DUET establishes a clearer and more robust forgetting boundary through the teacher's semantic understanding capability. This teacher-derived boundary, guided by contextualized instructions, is more interpretable and robust than boundaries defined directly by refusal response data, as the teacher can dynamically distinguish between queries targeting undesirable knowledge and those seeking general knowledge based on semantic comprehension rather than pattern matching.

Table 11: Overall results on the MUSE-Books (Harry Potter) benchmark: DUET delivers the most balanced unlearning performance. Methods with $\mathcal{D}_f^{\text{QA}}$ indicates that the forget set is the QA samples ($\mathcal{D}_f^{\text{query}} \cup \mathcal{D}_f^{\text{ans}}$) extracted from the raw book content; $\mathcal{D}_f^{\text{QR}} = \mathcal{D}_f^{\text{query}} \cup \mathcal{D}_f^{\text{refuse}}$ indicates a forget set of query-refusal response pairs (Sec 4.1). Methods without a data notation were trained on the raw book content. "+ KL" denotes a KL-divergence regularization augmented to minimize deviation from a reference model on the retention set $\mathcal{D}_r$.

| Method | R-Forget ↓ | R-Forget-500 ↓ | R-Retain ↑ | MMLU ↑ | Performance Shift ↑ |
|---|---|---|---|---|---|
| Base Model (Llama3.2-3B) | 32.13 | 39.99 | 84.29 | 61.46 | 0 |
| GA | 0.00 | 0.00 | 0.00 | 24.87 | -48.76 |
| GA + KL ($\mathcal{D}_r$) | 27.20 | 38.29 | 78.67 | 60.18 | -0.27 |
| GA ($\mathcal{D}_f^{\text{QA}}$) | 0.00 | 0.00 | 75.80 | 36.45 | 38.62 |
| GA ($\mathcal{D}_f^{\text{QA}}$) + KL ($\mathcal{D}_r$) | 27.44 | 36.87 | **84.95** | 60.62 | 7.63 |
| NPO | 24.18 | 26.83 | 69.69 | 54.79 | -0.16 |
| NPO + KL ($\mathcal{D}_r$) | 28.92 | 33.62 | 80.28 | 59.47 | 3.58 |
| NPO ($\mathcal{D}_f^{\text{QA}}$) | 30.19 | 34.28 | 46.20 | 60.48 | -31.42 |
| NPO ($\mathcal{D}_f^{\text{QA}}$) + KL ($\mathcal{D}_r$) | 21.55 | 25.60 | 26.38 | 60.55 | -33.85 |
| Refusal-Training ($\mathcal{D}_f^{\text{QR}} \cup \mathcal{D}_r$) | 31.02 | 37.75 | 75.32 | 60.48 | -6.60 |
| SimNPO | 17.60 | 21.41 | 43.09 | 60.40 | -9.15 |
| FLAT | **0.47** | **0.64** | 58.33 | 58.92 | 42.51 |
| SFR | 10.45 | 13.17 | 71.93 | 58.28 | 32.96 |
| UNDIAL | 28.20 | 29.09 | 73.92 | 58.08 | 1.08 |
| **DUET** ($\mathcal{D}_f^{\text{query}} \cup \mathcal{D}_r$) | 4.27 | 5.98 | 78.33 | **61.45** | **55.90** |

## A.6 DISTILLATION FROM GRADIENT ASCENT MODELS

We further examined whether distilling from an aggressively over-unlearned model could serve as an effective alternative teacher. To this end, we distilled DUET using a GA-unlearned model as the teacher. Two variants were evaluated: (i) distilling GA's behavior on both forget and retain sets, and (ii) distilling only GA's forget-set behavior without learning its logit distribution on retention data.

As shown in Table 12, distilling GA's full behavior led to poor performance, exhibiting both ineffective unlearning and substantial degradation of general knowledge, which mirroring GA's own catastrophic forgetting. To avoid inheriting GA's degraded retention behavior, we distilled only its forget-set logits. This retention-free variant improved forgetting but still lagged behind DUET across all metrics. These findings confirm that over-unlearned teachers with collapsed general utility are unsuitable for stable distillation, reinforcing the importance of DUET's in-context teacher design.

## A.7 GENERALIZATION ACROSS LLMS

To address the concern regarding limited evaluation, we conducted additional experiments on two widely representative LLMs: `mistralai/Mistral-7B-Instruct-v0.3` and `Qwen2.5-3B-Instruct`. We compared our method against the two top-performing baselines (Semi-NPO and FLAT) from our main experiments.

As shown in Tables 13 and 14, our method consistently outperforms these baselines across both models, demonstrating the strong generalizability of DUET across different LLM architectures.

## A.8 ABLATION ON MULTI-STEP DECODING DISTILLATION

We further examined whether extending unlearning distillation beyond the first decoding step could improve performance. Our design choice to distill only the initial token was motivated by efficiency and practicality. Although limited in depth, this approach still captures breadth through the top-$K$ candidate tokens, each of which can lead to valid unlearning trajectories in subsequent decoding. Following the reviewer's suggestion, we conducted ablation studies that distill to subsequent tokens $T \geq 1$. The unlearning objective for a training sample $x$ is formulated as:

$$\sum_{t=1}^{T} \ell(g_\theta(x \oplus y_{i<t}), g_{\text{ref}}(x_c \oplus y_{i<t})) \ \big| \forall 0 < t \leq T, \ y_t \sim \pi_{\text{ref}}(\cdot | x_c \oplus y_{i<t})$$

Table 12: Comparison of DUET with distillation from GA-based teachers on the MUSE-Books (Harry Potter) benchmark.

| Method | R-forget ↓ | R-forget_500 ↓ | R-retain ↑ | MMLU ↑ | Performance Shift ↑ |
|---|---|---|---|---|---|
| Base model | 32.13 | 39.99 | 84.29 | 61.46 | 0.00 |
| DUET | **4.27** | **4.86** | **78.33** | **61.52** | **57.09** |
| Distill from GA model | 17.40 | 19.54 | 42.86 | 54.53 | -13.18 |
| Distill from GA model (retain-free) | 18.04 | 20.32 | 77.82 | 57.55 | 23.38 |

Table 13: Performance comparison on Qwen2.5-3B-Instruct.

| Model (learning rate) | R-forget ↓ | R-forget_500 ↓ | knowmem_r ↑ | mmlu ↑ | Performance Shift ↑ |
|---|---|---|---|---|---|
| Qwen2.5 (base) | 30.04 | 31.89 | 73.39 | 65.48 | 0.00 |
| FLAT | 9.05 | 16.44 | 24.68 | 65.20 | -12.55 |
| Semi-NPO | 12.78 | 17.13 | 24.64 | 65.53 | -16.68 |
| DUET | 14.15 | 16.72 | 70.01 | 65.29 | **27.49** |

As shown in Table 15, rolling distillation over longer decoding horizons yields only marginal improvements in forgetting effectiveness, while causing noticeable degradation in general utility. These findings align with recent analyses of decoding dynamics Wang & Zhou (2024), which suggest that the first-step branching largely determines the global generation trajectory, whereas later positions mainly refine instead of redirect the model's behavior. This observation supports our conclusion that extending distillation beyond the initial step provides limited additional benefit and incurs extra utility cost.

We acknowledge that studying ability distillation for more complex reasoning tasks (e.g., multi-step mathematics or logical inference) remains an intriguing direction for future research. Nonetheless, our current one-step approach offers a balanced trade-off between unlearning efficacy and model capability preservation.

## A.9 ABLATION ON TRAINING DATA SIZE

We further examined the effect of training set size on unlearning performance. In response to this concern, we expanded our training data using GPT 5.1 with carefully designed prompts and scaled both the forget set and the retain set to 200 and 500 samples, respectively. We then conducted experiments with different hyperparameters on these expanded datasets.

**Results.** As shown in Table 16, the 200- and 500-sample configurations produce results highly comparable to our original 100-sample setup. This demonstrates that our model trained on 100 samples does not overfit and further validates the data efficiency of our approach. Moreover, these findings align with recent research on safety fine-tuning Pal et al. (2025); Liu et al. (2024b); Shi et al. (2024a); Pham et al. (2025), which indicates that data quality is more crucial than quantity, with 50–1000 carefully selected samples often sufficient to achieve effective safety refusal and efficient fine-tuning.

## A.10 ADVERSARIAL EVALUATION WITH JAILBREAK ATTACKS

To broaden our adversarial evaluation, we further tested DUET under a more sophisticated jailbreak technique Luo et al. (2025), which systematically reformulates explicitly harmful instructions into academic, exploratory, or hypothetically framed questions to conceal malicious intent.

Following the methodology in Luo et al. (2025), we created two categories of jailbreak prompts:

- **Educational**: raw requests framed as academic research (e.g., "for educational purposes").
- **Fanfic**: raw requests framed as creative writing needs (e.g., "if someone were writing fanfiction").

We applied these prompts to rewrite the 100-question forget set from MUSE. For each rewritten question, we queried the model 10 times and used an LLM-as-a-judge to determine whether the output revealed the target information. We report the attack success rate (ASR) for three methods in Table 17.

Table 14: Performance comparison on Mistral-7B-Instruct-v0.3.

| Model (learning rate) | R-forget ↓ | R-forget_500 ↓ | knowmem_r ↑ | mmlu ↑ | Performance Shift ↑ |
|---|---|---|---|---|---|
| Mistral_7B (base) | 48.40 | 50.40 | 89.71 | 59.74 | 0.00 |
| FLAT | 1.15 | 0.45 | 0.00 | 24.28 | -27.97 |
| Semi-NPO | 15.75 | 18.892 | 48.02 | 22.95 | -14.32 |
| DUET | 19.57 | 24.58 | 73.88 | 58.90 | **37.98** |

Table 15: Unlearning distillation with multi-step decoding.

| Method (learning rate) | R-forget ↓ | R-forget_500 ↓ | knowmem_r ↑ | mmlu ↑ | Performance Shift ↑ |
|---|---|---|---|---|---|
| base model | 32.13 | 39.99 | 84.29 | 61.46 | 0.00 |
| 1 token (3e-6)(original DUET) | 4.27 | 4.86 | 78.33 | 61.52 | **57.09** |
| 3 Token (1e-6) | 32.55 | 39.34 | 85.28 | 61.30 | 1.06 |
| 3 Token (3e-6) | 19.15 | 20.80 | 83.60 | 60.04 | 30.06 |
| 3 Token (6e-6) | 5.04 | 7.22 | 77.64 | 55.19 | **46.94** |
| 3 Token (1e-5) | 1.73 | 2.13 | 60.69 | 44.93 | 28.13 |
| Full (5e-7) | 32.03 | 37.65 | 84.29 | 61.24 | 2.22 |
| Full (1e-6) | 32.84 | 34.98 | 84.29 | 60.25 | 3.09 |
| Full (3e-6) | 5.86 | 6.12 | 77.28 | 55.86 | **47.53** |
| Full (6e-6) | 0.05 | 0.17 | 60.70 | 40.44 | 27.29 |

Table 16: Unlearning performance with varying data sizes.

| Method | R-forget ↓ | R-forget_500 ↓ | knowmem_r ↑ | mmlu ↑ | Performance Shift ↑ |
|---|---|---|---|---|---|
| base model | 32.13 | 39.99 | 84.29 | 61.46 | 0.00 |
| 100 (3e-6) | 4.27 | 4.86 | 78.33 | 61.52 | **57.09** |
| 200 (1e-6) | 33.06 | 31.37 | 79.95 | 59.17 | 1.06 |
| 200 (3e-6) | 4.02 | 5.81 | 76.66 | 59.13 | **52.33** |
| 200 (5e-6) | 2.06 | 1.52 | 39.50 | 51.42 | 13.71 |
| 500 (1e-6) | 31.20 | 30.26 | 80.66 | 59.08 | 4.65 |
| 500 (3e-6) | 4.57 | 5.51 | 76.49 | 59.35 | **52.13** |
| 500 (5e-6) | 3.08 | 3.12 | 54.42 | 49.48 | 24.07 |

While DUET achieves the lowest ASR among all compared approaches, it remains vulnerable to jailbreak attacks. This reveals a limitation of current unlearning methods focusing solely on objective formulation—such methods may struggle to maintain effective forgetting under adversarial prompting. This highlights an important direction for future research: improving the robustness of unlearned models against jailbreak attacks. Progress in this area will likely require advances in adversarial data curation, objective formulation, and comprehensive evaluation protocols.

Table 17: Evaluation under jailbreak attacks.

| Rank | Model | Fanfic | Educational | Overall ASR |
|---|---|---|---|---|
| 1st | DUET | 35.05% | 36.18% | **35.62%** |
| 2nd | FLAT | 44.26% | 30.79% | 37.53% |
| 3rd | GA | 45.25% | 34.26% | 39.76% |
| 4th | SimNPO | 49.80% | 38.02% | 43.91% |

## A.11 ABLATION ON PREFIX USAGE IN RETENTION DATA

We also investigated whether applying the unlearning prefix to the retention data in Eq. (3) is necessary. Specifically, we examined the effect of removing the teacher's prefix when processing the retention set. As shown in Table 18, the model's retention performance remains equally strong regardless of whether the prefix is applied to the retention data. This direct comparison under identical hyperparameters demonstrates negligible differences between the two configurations.

Table 18: Ablation study on prefix usage and retention strategies in DUET.

| Method | R-forget ↓ | R-forget_500 ↓ | knowmem_r ↑ | mmlu ↑ | Performance Shift ↑ |
|---|---|---|---|---|---|
| base model | 32.13 | 39.99 | 84.29 | 61.46 | 0.00 |
| original_du (with prefix in retain data) | 4.27 | 4.86 | 78.33 | 61.52 | **57.09** |
| du_retain_prefix_free | 4.71 | 5.37 | 77.82 | **59.16** | 53.27 |
| DUET + KL_retain | 28.33 | 30.33 | 81.09 | 59.04 | 7.84 |

We attribute this robustness to LLMs' strong semantic comprehension of user instructions. When the model receives queries unrelated to the target forgetting domain, it effectively ignores the unlearning prefix and responds normally to general-knowledge questions. Thus, the prefix does not adversely impact the model's ability to preserve general knowledge.

Additionally, this question motivated us to conduct further validation. We implemented an alternative configuration where DUET's retention component uses traditional KL-divergence to preserve general knowledge, rather than distillation with the prefix. As shown in the last row of Table 18, this approach significantly constrains DUET's forgetting effectiveness and prevents it from achieving the same level of performance. We attribute this to incompatibility between the boundary imposed by KL retention and the prefix-based supervision, which introduces conflicting training signals during optimization.

