# OpenReview forum: "DUET: Distilled LLM Unlearning from an Efficiently Contextualized Teacher"
_ICLR.cc/2026/Conference — ICLR 2026 Poster_

### Official Review · Reviewer_o7sB · 2025-10-27

**Soundness:** 4
**Presentation:** 3
**Contribution:** 3
**Rating:** 6
**Confidence:** 3

**Summary:**

The paper introduces DUET, a distillation-based LLM unlearning method that transfers the refusal behavior of an in-contextualized teacher into a student model via Top-K logit distillation. Concretely, the teacher is a frozen base LLM steered by a compact unlearning prefix; DUET trains the student to match the teacher’s dominant raw logit shifts on forget queries, while mixing in retain queries under the same objective to preserve utility. DUET requires only query-level forget data without no ground-truth answers or refusal templates, and achieves strong forgetting–retention trade-offs on LLM unlearning benchmarks. Comprehensive ablations also indicate robustness to reverse-prompt attacks and evaluation-format shifts.

**Strengths:**

1. The paper is well written and easy to follow.

2. Distilling from an in-context teacher to obtain an unlearned student is reasonable, cleanly formalized, and avoids constructing ground-truth answers or refusal templates for forget data.

3. The work proposes and validates the effectiveness of Top-K logit distillation for unlearning.

4. The experimental section demonstrates the method’s effectiveness and provides thorough ablations showing robustness to reverse-prompt attacks and evaluation-format shifts.

**Weaknesses:**

The paper’s discussion of distillation-based unlearning remains incomplete. Similar ideas have been applied in the LLM unlearning literature, including W2SDefense (weak-to-strong distillation for backdoor removal) [1], UNDIAL (self-distillation with adjusted logits) [2], and UNDO (“distillation robustifies unlearning”) [3]. The manuscript should explicitly discuss conceptual and practical differences from these works, and add an analysis to clarify DUET’s unique contributions.

### Reference

1. Zhao, Shuai et al. “Unlearning Backdoor Attacks for LLMs with Weak-to-Strong Knowledge Distillation.” ACL (2024).

2. Dong, Yijiang River et al. “UNDIAL: Self-Distillation with Adjusted Logits for Robust Unlearning in Large Language Models.” NAACL (2024).

3. Lee, Bruce W. et al. “Distillation Robustifies Unlearning.” ArXiv.

**Questions:**

1. The paper states that the Harry Potter (MUSE-Books) evaluation set is expanded to 500 items. How exactly is this expansion performed?

2. Please clarify the source and selection process for the retention data used during training and evaluation.

3. In Table 1, why does NPO with a retain-set KL (w $\text{KL}(\mathcal{D_r})$, line 320) yield worse retain performance than the variant without retain-set KL (w/o $\text{KL}(\mathcal{D_r})$, line 319)? Please explain the underlying cause.

---

> ### Author Response · Authors · 2025-11-24
> **Response to Weaknesses**
>
> Thank you for the thoughtful and constructive review. We appreciate your careful analysis, especially regarding related distillation-based unlearning work and the KL-retain behavior. Your comments were highly helpful, and we address each point in detail below.
>
> ---
>
> ## Weaknesses:
> ### W1: discussion of distillation-based unlearning remains incomplete.
>
> **Response**: Thank you for rasing this important point and for directing us to related work [1-3]. While conceptually related as distillation-based defenses, W2SDefense [1] and UNDO [3] operate under substantially different threat models and are therefore not suitable as direct baselines for DUET.
>
> W2SDefense [1] targets backdoor removal in poisoned LLMs by training a separate clean teacher and distilling it into the compromised model to suppress trigger behavior, whereas UNDO [3] assumes an existing unlearning algorithm has already been applied and then uses distillation only as a post-hoc mechanism to make the unlearned model more robust to subsequent "relearning" under further fine-tuning. In contrast, DUET addresses targeted open-domain knowledge removal from a non-poisoned model and uses distillation itself as the core unlearning mechanism, rather than as an auxiliary defense layer around another unlearning procedure.
>
> By comparison, UNDIAL [2] is much closer to our setting: it is also a distillation-based unlearning approach. However, UNDIAL removes targeted knowledge by directly shifting the model's logits toward predefined refusal tokens, rather than mimicking an in-context teacher. We therefore include it as a baseline to compare against our teacher-guided logit-distillation framework. The results are shown below:
>
>
> | Method (learning rate) | R-forget ↓ | R-forget_500 ↓ | knowmem_r ↑ | mmlu ↑ | Performance Shift ↑ |
> | --- | --- | --- | --- | --- | --- |
> | base model | 32.13 | 39.99 | 84.29 | 61.46 | 0.00 |
> | DUET (3e-6) | 4.27 | 4.86 | 78.33 | 61.52 | **57.09** |
> |
> | undial (3e-6) | 37.57 | 38.36 | 78.28 | 60.25 | -11.03 |
> | undial (6e-6)  | 28.20 | 29.09 | 73.92 | 58.08 | **1.08** |
> | undial (1e-5)  | 22.20 | 24.55 | 55.95 | 52.68 | -11.75 |
>
> **Results**: As shown in **Table 1**, UNDIAL underperforms compared to our method. We attribute this to several key differences: First, UNDIAL optimizes for Memorization Accuracy (MA) and Extraction Likelihood (EL), which target different forgetting objectives than ours. Second, UNDIAL's teacher model for forgetting and retaining knowledge follows the same paradigm as other training-based methods—it directly suppresses target knowledge outputs, which does not fundamentally differ from existing approaches in its core mechanism.
>
> In contrast, we believe DUET establishes a clearer and more robust forgetting boundary through the teacher's semantic understanding capability. This teacher-derived boundary, guided by contextualized instructions, is more interpretable and robust than boundaries defined directly by refusal response data, as the teacher can dynamically distinguish between queries targeting undesirable knowledge and those seeking general knowledge based on semantic comprehension rather than pattern matching.
>
> ---
>
> References:
>
> [1] Zhao, Shuai et al. “Unlearning Backdoor Attacks for LLMs with Weak-to-Strong Knowledge Distillation.” ACL (2024).
>
> [2] Dong, Yijiang River et al. “UNDIAL: Self-Distillation with Adjusted Logits for Robust Unlearning in Large Language Models.” NAACL (2024).
>
> [3] Lee, Bruce W. et al. “Distillation Robustifies Unlearning.” ArXiv.

---

> ### Author Response · Authors · 2025-11-24
> **Response to Questions**
>
> ### Q1: How exactly is the evaluation set expansion performed?
> **Response**: We expanded the evaluation set by prompting an LLM (GPT-4o) to generate additional samples. Due to space constraints, we have included the complete prompts in our open-source code repository. In brief, we instructed the LLM to generate 400 question-answer pairs based on the content of all seven Harry Potter books and the original 100 test samples. The generation criteria required that: (1) answers should be direct and straightforward, (2) answers should have a unique correspondence to questions, (3) questions should be simple and explicit, and (4) the question types should be diverse and uniformly distributed across all seven books.
>
> ---
>
> ### Q2 Clarify the source and selection process for the retention data.
> **Response**: Similarly, our retention test data consists of simple questions with ground-truth answers generated by GPT-4o, sampled from Wikipedia. The MUSE [1] retain set was constructed using a comparable methodology. We have also released our prompts in the open-source repository. The generation instructions require the LLM to create questions that are: well-known, common sense knowledge. After generation, we performed verification by invoking another LLM to validate the correctness of the answers, which achieved 100% accuracy. Therefore, we believe this dataset qualifies as a robust general knowledge evaluation set.
>
> ---
>
> ### Q3 Why does NPO perform worse when KL-retention regularization is applied?
> **Response**: Thank you for this astute observation. We attribute this phenomenon to the relatively small size of the retention data combined with the strong constraint imposed by the KL-divergence term. This combination leads to overfitting during training, creating conflicts with the forgetting loss term. Consequently, this damages the model's overall question-answering capability, simultaneously hurting both unlearning effectiveness and general knowledge performance as measured by ROUGE scores.
>
> ---
>
> Reference:
>
> [1] Shi, Weijia et al. "MUSE: Machine Unlearning Six-Way Evaluation for Language Models." ICLR (2025).
>
> ---
>
> **Thank you again for your valuable comments and constructive suggestions.** We hope that our response with additional experiments can earn your favorable consideration.

---

> > ### Comment · Reviewer_o7sB · 2025-11-26
> >
> > Thank you for the author's reply. Most of my concerns have been resolved. The added baselines strongly support the method's advantages. I have raised my score.

---

> > > ### Author Response · Authors · 2025-11-26
> > > **Thank you!**
> > >
> > > Thank you for your follow-up and for reconsidering the score! We appreciate your careful evaluation and are glad that our response addressed your concerns.

---

### Official Review · Reviewer_BGFT · 2025-10-27

**Soundness:** 3
**Presentation:** 3
**Contribution:** 3
**Rating:** 4
**Confidence:** 3

**Summary:**

The paper proposes **DUET**, a distillation-based unlearning framework that transfers a teacher model’s refusal behavior into a student model via top-K logit alignment at the first decoding step. DUET aims to retain general capabilities while suppressing undesirable knowledge. Experiments on MUSE-Books and WMDP show strong forgetting with comparatively small utility loss, and improved robustness to a simple reverse-prompt attack.

**Strengths:**

1. **Simple method and clear motivation.** The approach is conceptually straightforward and is well justified relative to tuning-based and purely in-context unlearning.
2. **Good empirical performance.** On MUSE-Books and WMDP, DUET achieves lower forgetting scores and stronger utility preservation than most baselines.
3. **Robustness to attack.** The distilled model is notably less sensitive to a reverse-prompt attack than a purely in-context teacher.

**Weaknesses:**

1. **Distilling only the first decoding step is not fully convincing.** Many tasks (e.g., math reasoning) often start with stereotyped lead tokens (e.g., _“To solve the problem, I need to…”_), so aligning only the first-step logits may fail to shape downstream generation in a robust way. The paper explicitly trains on **first-position** logits only; a deeper justification and multi-step ablations would help.
2. **Limited experimental breadth.** The forget set and retention set used for training are quite small (e.g.,  100 queries for Harry-Potter forgetting; 100 for retention), which risks overfitting the teacher prefix distribution and may bias results toward DUET’s design. Broader forget/retention sets or more domains would strengthen claims.
3. **Adversarial evaluation is narrow.** “Reverse attacks” are instantiated as a single reverse-prompt. The paper does not cover more systematic jailbreak or targeted relearning attack suites, so robustness claims remain preliminary.

Overall, I like this paper, and if the authors can provide a clear explanation about W1 and W2, I’d be happy to raise my score.

**Questions:**

1. Why apply the unlearning prefix to _retention_ data in Eq. (3)? A seemingly cleaner design is to prefix only the forget set $D_f$​ and leave $D_r$ unmodified.
2. Is this aggregate metric "performance shift" standard in the unlearning literature, or specific to this paper?

---

> ### Author Response · Authors · 2025-11-24
> **Response to Weakness 1 and 2**
>
> Thank you for your valuable and constructive review. We are highly encouraged that you like our paper! Please find below our detailed responses to your comments.
>
> ---
> ### W1 Distilling only the first decoding step is not fully convincing.
> **Response**: We appreciate this insightful comment. Our design choice to distill only the first decoding step was motivated by efficiency and practicality. While limited in depth, this approach still captures breadth by considering the top-K candidate tokens, each of which has the potential to lead to valid unlearning responses in subsequent decoding. Following your  suggestion, we conducted ablation studies extending distillation to subsequent tokens $T \geq 1$, where the unlearning objective for a training sample $x$  is formulated as blow:
>
> $\sum_{t=1}^{T}  l(g_\theta(x \oplus y_{i<t}), g_\text{ref}(x_\text{ic}  \oplus  x  \oplus  y_{i < t}))| \forall 0 <t \leq T, {y_t \sim \pi_\text{ref}(\cdot|x_\text{ic}  \oplus  x  \oplus y_{i < t} )}$
>
> **Results**: Results in **Table 1** show that rolling unlearning distillation to longer token sequences yields marginal improvements in forgetting effectiveness, but causes noticeable degradation in model utility performance.
>
> We further note that recent analyses of decoding dynamics (**paper [1]**) show that the first-step branching largely determines the global generation trajectory, while later positions mainly refine rather than redirect the model’s behavior. This supports our finding that extending distillation beyond the initial step offers limited additional benefit but incurs extra utility cost.
>
> We acknowledge that for ability distillation on more complex reasoning tasks (*e.g.*, math problems), exploring higher breadth (top-$K$ candidates) and depth (token index $T$) represents an intriguing direction for future work. In contrast, our current one-step approach presents a balanced trade-off between unlearning efficacy and model capabilities.
>
> **Table 1: Unlearning distillation with multi-step decoding.**
>
> | Method (learning rate) | R-forget ↓ | R-forget_500 ↓ | knowmem_r ↑ | mmlu ↑ | Performance Shift ↑ |
> | --- | --- | --- | --- | --- | --- |
> | base model | 32.13  | 39.99 | 84.29  | 61.46  | 0.00 |
> | 1 token (3e-6) | 4.27 | 4.86 | 78.33 | 61.52 | **57.09** |
> | 3 Token (1e-6) | 32.55 | 39.34 | 85.28 | 61.30 | 1.06 |
> | 3 Token (3e-6) | 19.15 | 20.80 | 83.60 | 60.04 | 30.06 |
> | 3 Token (6e-6) | 5.04 | 7.22 | 77.64 | 55.19 | **46.94** |
> | 3 Token (1e-5) | 1.73 | 2.13 | 60.69 | 44.93 | 28.13 |
> | Full (5e-7) | 32.03 | 37.65 | 84.29 | 61.24 | 2.22 |
> | Full (1e-6) | 32.84 | 34.98 | 84.29 | 60.25 | 3.09 |
> | Full (3e-6) | 5.86 | 6.12 | 77.28 | 55.86 | **47.53** |
> | Full (6e-6) | 0.05 | 0.17 | 60.70 | 40.44 | 27.29 |
>
> ---
> ### W2 Limited experimental breadth.
> **Response**: We  appreciate this valid concern regarding the training set size. In response,  we expanded our training data using GPT 5.1 with carefully designed prompts and scaled both the forget set and retain set to 200 and 500 samples, respectively. We then conducted experiments with different hyperparameters on these expanded datasets.
>
> **Results**: As shown in **Table 2** below, the 200/500-sample configurations produce results highly comparable to our original 100-sample setup. This demonstrates that our model trained on 100 samples does not overfit, and further validates the data efficiency of our approach.
>
> Moreover, our results align with  findings in recent  research on safety fine-tuning [2-5], which indicate that data quality is more crucial than quantity, with 50-1000 carefully selected samples sufficient to achieve effective safety refusal and efficient fine-tuning.
>
> **Table 2: Unlearning with varying data size.**
>
> | Method | R-forget ↓ | R-forget_500 ↓ | knowmem_r ↑ | mmlu ↑ | Performance Shift ↑ |
> | --- | --- | --- | --- | --- | --- |
> | base model | 32.13  | 39.99 | 84.29  | 61.46  | 0.00 |
> | 100 (3e-6) | 4.27 | 4.86 | 78.33 | 61.52 | **57.09** |
> | 200_1e-6 | 33.06 | 31.37% | 79.95 | 59.17 | 1.06 |
> | 200_3e-6 | 4.02 | 5.81% | 76.66 | 59.13 | **52.33** |
> | 200_5e-6 | 2.06 | 1.52% | 39.50 | 51.42 | 13.71 |
> | 500_1e-6 | 31.20 | 30.26% | 80.66 | 59.08 | 4.65 |
> | 500_3e-6 | 4.57 | 5.51% | 76.49 | 59.35 | **52.13** |
> | 500_5e-6 | 3.08 | 3.12% | 54.42 | 49.48 | 24.07 |
>
> References:
>
> [1] Wang, Xuezhi et al. “Chain-of-Thought Reasoning Without Prompting.” NeurIPS (2024).
>
> [2] Pal, Soumyadeep et al. “LLM Unlearning Reveals a Stronger-Than-Expected Coreset Effect in Current Benchmarks.” arXiv preprint (2025).
>
> [3] Liu, Yilun et al. “CoachLM: Automatic Instruction Revisions Improve the Data Quality in LLM Instruction Tuning.” ICDE (2024).
>
> [4] Shi, Taiwei et al. “Safer-Instruct: Aligning Language Models with Automated Preference Data.” NAACL (2024).
>
> [5] Pham, Anh et al. “Preventing Catastrophic Forgetting: Behavior-Aware Sampling for Safer Language Model Fine-Tuning.” arXiv (2025).

---

> ### Author Response · Authors · 2025-11-24
> **Response to Weakness 3 and Questions**
>
> ### W.3 Adversarial evaluation is narrow.
> **Response**: Thank you for this valuable feedback. In response, we evaluated our method against a more sophisticated jailbreak technique [1] (Luo, Xuan, et al. "A Simple and Efficient Jailbreak Method Exploiting LLMs' Helpfulness." 2025.) that systematically reformulates explicitly harmful instructions into "academic, exploratory, and hypothetically-framed" questions to hide true intent.
>
> Following the methodology in [1], we created two types of jailbreak prompts:
>
> - **Educational**: we framed a raw request as academic research (e.g., "for educational purposes")
> - **Fanfic**: we framed raw requests as for creative writing needs (e.g., "if someone were writing fanfiction")
>
> We applied these prompts to rewrite the 100-question forget set from MUSE. For each question, we repeated query for 10 times and used an LLM-as-a-judge to determine whether the output revealed the target information. We report the attack success rate (ASR) across three methods in **Table 3** below.
>
> While our method (DUET) achieves the lowest ASR among all compared approaches, it remains vulnerable to jailbreak attacks. This reveals a limitation that current unlearning methods focusing on objective formulation may struggle to maintain effective forgetting under adversarial prompting. This highlights a critical direction for future research: improving the robustness of unlearned models against jailbreak attacks. Achieving this will likely require advances in multiple areas, including curation of  adversarial training dataset, objectives formulation, and comprehensive evaluation protocols.
>
> **Table3: Evaluation under jailbreak attacks.**
>
> | Rank | Model | Fanfic | Educational | Overall ASR |
> | --- | --- | --- | --- | --- |
> | 🥇 1st | DUET | 35.05% | 36.18% | **35.62%** |
> | 🥈 2nd | FLAT | 44.26% | 30.79% | **37.53%** |
> | 🥉 3rd | GA | 45.25% | 34.26% | **39.76%** |
> |  4th | SimNPO | 49.80% | 38.02% | **43.91%** |
>
> ---
> ## Questions:
>
> ### Q1. Why apply the unlearning prefix to retention data in Eq. (3)?
> Thank you for this careful observation. We did consider removing the teacher's prefix when processing the retention set. In fact, we conducted experiments on this exact question. As shown in **Table 4** below, we found that the model's retention performance remains equally strong regardless of whether the prefix is applied to retention data. The table presents a direct comparison under identical hyperparameters, demonstrating negligible differences between the two configurations.
>
> **Table 4: Ablation study on prefix usage and retention strategies in DUET**
> | Method | R-forget ↓ | R-forget_500 ↓ | knowmem_r ↑ | mmlu ↑ | Performance Shift ↑ |
> | --- | --- | --- | --- | --- | --- |
> | base model | 32.13 | 39.99 | 84.29 | 61.46 | 0.00 |
> | original_du(w/ prefix in retain data) | 4.27 | 4.86 | 78.33 | 61.52 | 57.09 |
> | du_retain_prefix_free | 4.71 | 5.37 | 77.82 | **59.16** | **53.27** |
> | DUET+kl_retain | 28.33 | 30.33 | 81.09 | 59.04 | 7.84 |
>
> We attribute this robustness to LLMs' strong semantic comprehension of user instructions. When the model encounters queries unrelated to the target forgetting domain, it effectively ignores the unlearning prefix and responds normally to general knowledge questions. Thus, we conclude that the prefix does not adversely impact the model's ability to preserve general knowledge.
>
> Additionally, your question motivated us to conduct further validation. We implemented an alternative where DUET's retention component uses traditional KL-divergence to preserve general knowledge, rather than distillation with the prefix. As shown in the last row of **Table 4**, this approach significantly constrains DUET's forgetting effectiveness, preventing it from achieving the same level of performance. We attribute this to incompatibility between the boundary imposed by KL retention and the prefix-based supervision, which creates conflicting training signals during optimization.
>
> ---
>
> ### Q2 Is this aggregate metric "performance shift" standard in the unlearning literature, or specific to this paper?
>
> Aggregating metric changes into a unified performance indicator is a widely adopted approach in the unlearning literature, as evidenced by recent works [1-3]. While our formulation incorporates a slightly larger number of component metrics to provide more comprehensive coverage, the fundamental methodology aligns with established practices in the field.
>
>
> ---
>
> References:
>
> [1] Wang, Yaxuan et al. "LLM Unlearning via Loss Adjustment with Only Forget Data." ICLR (2025).
>
> [2] Maini, Pratyush et al. "TOFU: A Task of Fictitious Unlearning for LLMs." COLM (2024).
>
> [3] Deng, Zhijie et al. "GUARD: Generation-time LLM Unlearning via Adaptive Restriction and Detection." arXiv (2025).
>
> ---
>
> **Thank you again for your valuable comments and constructive suggestions.** We hope that our response with additional experiments can earn your favorable consideration.

---

> > ### Comment · Reviewer_BGFT · 2025-11-25
> >
> > Thanks for your detailed response and additional experiments. Although the design of some method components still puzzles me, the author's ablation experiments, to some extent, demonstrate the effectiveness of the design. Considering the innovativeness of the method, I have raised my score. And I hope the authors can supplement the experimental results from the rebuttal in the revision.

---

> > > ### Author Response · Authors · 2025-11-26
> > > **Thank you!**
> > >
> > > Thank you for the thoughtful follow-up and for raising the score! We appreciate your careful assessment, and we are glad the additional experiments helped clarify our design. We have already incorporated the updated experimental results into the revision.

---

### Official Review · Reviewer_Vd1V · 2025-11-01

**Soundness:** 2
**Presentation:** 2
**Contribution:** 2
**Rating:** 4
**Confidence:** 3

**Summary:**

This paper introduces DUET (Distilled Unlearning from an Efficient Teacher), a novel distillation-based method for large language model (LLM) unlearning—the process of removing undesirable knowledge without full retraining. Existing unlearning techniques either suffer from high computational cost and catastrophic forgetting (tuning-based approaches) or security vulnerabilities such as prompt removal and reverse-engineering attacks (in-context methods). DUET addresses these issues by training a student model to emulate a prompt-steered teacher that suppresses unwanted knowledge while retaining useful domain capabilities. Experiments on benchmark datasets show that DUET delivers forgetting and utility preservation, achieving good performance with higher data efficiency than state-of-the-art methods.

**Strengths:**

The novelty of DUET lies in its distillation-based unlearning approach, where a student model learns from a prompt-steered teacher to selectively suppress undesirable knowledge while preserving useful information. This method combines efficiency, robustness, and effectiveness, outperforming existing unlearning techniques in both forgetting unwanted content and maintaining model utility.

The research provides:

•	Efficient unlearning without full retraining, saving computational resources.

•	Effective knowledge removal while preserving useful domain information.

•	High data efficiency compared to existing unlearning methods, achieving reasonable performance on benchmarks.

**Weaknesses:**

The research demonstrates some shortcomings

•	Reliance on teacher quality — form my understanding effectiveness depends on how well the teacher model suppresses unwanted knowledge.

•	The work makes a valuable contribution and builds effectively on current advances. However, including a discussion of remaining challenges and possible avenues for future research would strengthen the paper and highlight its long-term potential.

•	Limited evaluation — generalisation across LLMs is not shown.

**Questions:**

Please check your references – there are titles that have some “words” like LLM that should be uppercase.

With regard to the use of LLMs, I suggest that you check the claims that are made, such as “comprehensive evaluations” and “significantly superior performance”. It’s up to the reader to judge if the evaluations are comprehensive and whether the performance is “improved” or “significantly superior”. Please discuss.

Please discuss the generalizability of the results across LLMs.

---

> ### Author Response · Authors · 2025-11-24
> **Response to Weaknesses 1 and 2**
>
> Thank you for your valuable and constructive review. We are encouraged by your recognition of the novelty and efficacy of our method. Please find below our responses to your comments.
>
> ---
>
> ### W1: Reliance on teacher quality.
> Thank you for this insightful comment. We'd like to address this from three perspectives:
>
> 1) while teacher quality is indeed a decisive factor, it serves as an upper bound rather than a limitation. In fact, all supervised learning approaches have upper bounds (e.g., SFT data quality) that determine the learning direction.
> 2) Our experiments use a high-quality teacher model that effectively refuses to answer forget-set questions (see **Table 6**, unlearning performance w/o Reverse Attack for the Base model with prefix). Moreover, as LLMs evolve with improved semantic understanding and prompt-following capabilities, this upper bound will continously increase for current and future unlearning research.
> 3) Finally, unlike static datasets (e.g. as in refusal training), the latent logit distribution from the teacher model serves as a more refined guidance based on its prompt understanding, which offers more effective guidance for unlearning than directly training on output samples.
>
> ---
> ### W2. The work makes a valuable contribution and builds effectively on current advances but lacks discussion of remaining challenges and possible avenues for future research.
>
> **Response**: We thank the reviewer for this valuable suggestion. We are pleased to expand our discussion on future challenges and research directions related to this work. The manuscript has been updated to discuss the following open challenges for future research:
>
> * **Unlearning Boundary Determination**: We believe that determining the boundary of unlearning represents a significant challenge, specifically regarding *what to forget versus what to retain*. Current unlearning methods, particularly training-based approaches where boundaries are defined through data face specific challenges in this field. This creates substantial problems: when the distinction between forget data and retain data is ambiguous, samples near the boundary exhibit poor unlearning performance. This manifests as the most prominent trade-off between unlearning and retaining. To address the boundary determination problem, our method offers a distinct perspective by leveraging prompt-based steering and the LLM's semantic understanding capabilities. This enables the model to autonomously judge whether a query pertains to sensitive knowledge, allowing us to refine boundaries through carefully crafted prompts. Our work provides a promising  solution to this challenge, and we anticipate future work will develop more sophisticated methods to resolve the boundary ambiguity.
>
> * **Evaluation Protocol**: How to evaluate whether a model has truly unlearned is another critical challenge. Due to the nature of unlearning, we cannot simply assess whether the model has forgotten knowledge based solely on its surface-level responses. We must also examine whether its response patterns regarding forgotten knowledge have fundamentally changed. Additionally, it is difficult to evaluate whether the model merely refuses to answer on the surface while still retaining deep knowledge of the unlearning target. Existing methods, including Membership Inference Attacks (MIA) and jailbreak techniques, remain insufficient for comprehensively evaluating LLM unlearning, as models can still evade detection. Our work makes meaningful contributions in this aspect: we expanded the original 100-question forget testset to 500 questions, covering a more comprehensive and diverse range of question types and domains. The quality of answers has also been enhanced. For example, we strictly enforce that answers uniquely and one-to-one correspond to questions, are widely known, and have been carefully verified.

---

> ### Author Response · Authors · 2025-11-24
> **Response to Weakness 3 and Questions**
>
> ### W3/Q3: Limited evaluation — generalisation across LLMs is not shown.
>
> **Response**: Thank you for pointing this out. To address this valid concern, we conducted additional experiments on two widely-representative LLMs: mistralai/Mistral-7B-Instruct-v0.3 and Qwen2.5-3B-Instruct. We compared our method against the two top-performing baselines (Semi-NPO and FLAT) from our main experiments. As shown in **Table 1 and 2** below, our method consistently outperforms these baselines across both models, demonstrating the strong generalizability of DUET across different LLMs.
>
> **Table 1: Performance comparison on Qwen2.5-3B-Instruct.**
>
> | Model (learning rate) | R-forget↓ | R-forget_500↓ | knowmem_r↑ | mmlu↑ | Performance Shift↑ |
> | --- | --- | --- | --- | --- | --- |
> | Qwen2.5 (base) | 30.04 | 31.89 | 73.39 | 65.48 | 0.00 |
> | FLAT | 9.05 | 16.44 | 24.68 | 65.20 | -12.55 |
> | Semi-NPO | 12.78 | 17.13 | 24.64 | 65.53 | -16.68 |
> |
> | DUET (5e-7) | 17.86 | 20.45 | 72.12 | 65.38 | 22.25 |
> | DUET (1e-6) | 16.24 | 19.93 | 71.09 | 65.30 | 23.28 |
> | DUET (3e-6) | 14.15 | 16.72 | 70.01 | 65.29 | **27.49** |
> | DUET (6e-6) | 12.37 | 13.15 | 62.05 | 65.13 | 24.72 |
> | DUET (1e-5) | 9.10 | 13.21 | 40.47 | 59.81 | 1.03 |
>
> ---
>
> **Table 2: Performance comparison on Mistral-7B-Instruct-v0.3.**
>
> | Model (learning rate) | R-forget↓ | R-forget_500↓ | knowmem_r↑ | mmlu↑ | Performance Shift↑ |
> | --- | --- | --- | --- | --- | --- |
> | Mistral_7B (base) | 48.40 | 50.40 | 89.71 | 59.74 | 0.00 |
> | FLAT | 1.15 | 0.45 | 0.00 | 24.28 | -27.97 |
> | Semi-NPO | 15.75 | 18.892 | 48.02 | 22.95 | -14.32 |
> |
> | DUET (5e-7) | 43.37 | 48.04 | 86.75 | 59.72 | 4.41 |
> | DUET (1e-6) | 40.21 | 46.68 | 85.58 | 59.64 | 7.68 |
> | DUET (2e-6) | 30.44 | 32.88 | 80.61 | 59.25 | 25.89 |
> | DUET (3e-6) | 19.57 | 24.58 | 73.88 | 58.90 | **37.98** |
> | DUET (4e-6) | 0.96 | 1.06 | 13.46 | 24.92 | -14.29 |
> | DUET (5e-5) | 0.50 | 0.46 | 0.24 | 22.97 | -28.40 |
>
> ---
>
> ## Questions: Please check your references and claims
> **Response:** Thank you for your careful review. Regarding your suggestions about our references, we have made the necessary corrections. We have also revised our descriptions of LLM performance throughout the paper to adopt a more objective tone. For example, we changed "significantly superior performance" to "higher performance." Please see the revised PDF for full details.
>
> ---
>
> **Thank you again for your valuable comments and constructive suggestions.** We hope that our response with additional experiments can earn your favorable consideration.

---

> > ### Comment · Reviewer_Vd1V · 2025-11-25
> >
> > Thanks for your detailed response. I particularly appreciated the carefully considered response around challenges. This demonstrated a deep understanding of the problem and solution. I have raised my score and look forward to see the additional material from the rebuttal  in the revision.

---

> > > ### Author Response · Authors · 2025-11-26
> > > **Thank you!**
> > >
> > > Thank you for the thoughtful feedback and for raising the score! We appreciate your recognition of the challenges and our responses. The additional material from the rebuttal has already been integrated into the revision.

---

### Official Review · Reviewer_47LM · 2025-11-01

**Soundness:** 3
**Presentation:** 3
**Contribution:** 3
**Rating:** 6
**Confidence:** 3

**Summary:**

This paper proposes DUET, a novel distillation-based unlearning method that combines the merits of two approaches: optimizing a student model to imitate the behavior of a prompt-steered teacher, which effectively refuses the generation of undesirable knowledge while preserving general domain knowledge. Experiments on existing benchmarks demonstrate that DUET achieves strong performance in both forgetting and utility preservation, with greater data efficiency.

**Strengths:**

1. The proposed approach is novel and sound. The paper reveals that training-based unlearning achieves stronger robustness but risks greater utility degradation, while contextualized unlearning enables more precise unlearning yet can be easily reversed. The proposed method strikes a good balance between these two paradigms.
2. The paper proposes top-k logits distillation to further enhance performance.
3. The paper is enjoyable to read.

**Weaknesses:**

The authors should compare their method with more baseline methods, such as [1][2][3], as well as additional distillation-based approaches—for instance, distillation from gradient ascent methods.

How about distillation from multiple unlearning teachers?

[1] Unified Gradient-Based Machine Unlearning with Remain Geometry Enhancement, NeurIPS'24
[2] Salun: Empowering machine unlearning via gradient-based weight saliency in both image classification and generation, ICLR'24
[3] Torward Natural Machine Unlearning, TPAMI'25

**Questions:**

see weakness

---

> ### Author Response · Authors · 2025-11-24
> **Response to Weakness 1**
>
> Thank you for your thoughtful and constructive feedback. We appreciate your valuable suggestions for strengthening our work through additional baseline comparisons. We have conducted extensive additional experiments to address your concerns. Please find below our detailed responses to each of your comments.
> ## Weaknesses:
>
> ### W1.1: Authors should compare with more baseline methods, such as [1][2][3]:
> **Response**: Thank you for your valuable suggestions and the recommended papers. After carefully reviewing all three articles, we believe the first **paper [1]** is most appropriate for comparison. Here are our considerations for each:
>
> We respectfully believe that **papers [2, 3]** are not suitable baselines for our work. **Paper [2]** is designed specifically for CV unlearning, and whether its core concept (parameter saliency for images) can be transferred to NLP tasks lacks theoretical or empirical support . **Paper [3]** fuses unlearning set data with retain set data from incorrect classes at the *pixel-level* level. Adapting this idea to the *token level* fusion in NLP would require substantial modifications and remains unvalidated in the literature.
>
> **Paper [1]** (SFR) performs machine unlearning by following a Hessian-guided forgetting direction on a remain-preserving manifold. Although originally designed for CV, its methodology is well-grounded to be adapted to NLP  settings. We have added experiments comparing this method to ours on the WMDP and MUSE benchmarks. As shown in **Table 1**, it does not outperform our approach and shows catastrophic forgetting on general knowledge. *Please note: Since the paper did not provide code for NLP settings, we reimplemented the method based on our best understanding.*
>
> **Table 1: Comparing DUET with SFR ([1]).**
>
> | Method (learning rate) | R-forget ↓ | R-forget_500 ↓ | knowmem_r ↑  | mmlu ↑ | Performance Shift ↑ |
> | --- | --- | --- | --- | --- | --- |
> | base model | 32.13 | 39.99 | 84.29 | 61.46 | 0.00 |
> | DUET (3e-6) | **4.27** | **4.86** | **78.33** | **61.52** | **57.09** |
> |
> |SFR (1e-6) | 32.71 | 37.89 | 82.93 | 61.51 | 0.21|
> | SFR (3e-6) | 10.45 | 13.17 | 71.93 | 58.28 | **32.96**|
> | SFR (6e-6) | 0.07 | 0.03 | 10.19 | 28.32 | −35.22|
>
>
> In addition to the three papers you mentioned, Reviewer o7sB also suggested that we consider the **paper [4]** (UNDIAL). It pushes the model’s logits toward refusal tokens rather than imitating an in-context teacher, making it a suitable baseline for our method. For the experimental results, please refer to our response to Reviewer o7sB.
>
> References:
>
> [1] Huang Zhehao et al. “Unified Gradient-Based Machine Unlearning with Remain Geometry Enhancement.” NeurIPS (2024).
>
> [2] Fan Chongyu et al. “SalUn: Empowering Machine Unlearning via Gradient-based Weight Saliency in Both Image Classification and Generation.” ICLR (2024).
>
> [3] He Zhengbao et al. “Towards Natural Machine Unlearning.” IEEE TPAMI (2025).
>
> [4] Dong Yijiang River et al. “UNDIAL: Self-Distillation with Adjusted Logits for Robust Unlearning in Large Language Models.” NAACL (2025).
>
>
>
> ---
> ### W1.2: Authors should compare with additional distillation-based approaches—for instance, distillation from gradient ascent methods.
>
> **Response**: Thank you for this intriguing suggestion! We believe distilling from models with weaker unlearning performance would be ineffective, as their upper bound is already lower than ours. However, we identified a potentially promising scenario: distilling from an **over-unlearned** model might preserve strong unlearning. Thus, we selected GA that achieves strong forgetting effects as the distillation teacher.
>
> **Results:** As shown in **Table 2**,  distilling GA's behavior on both forget and retain sets leads to poor overall performance, with ineffective unlearning and notably dropped performance on the general knowledge. This is likely becuase that we inherited GA's catastrophic forgetting characteristics. ***So we thought of abandoning the inheritance of GA's performance in general knowledge.*** When distilling **only** on GA's forget-set behavior without learning its logit distribution on retention data (*i.e.* retention free), the unlearning efficacy improved but still underperformed our method.
>
> **Table 2: Comparing DUET with GA-as-Teacher:**
>
> | Method | R-forget ↓ | R-forget_500 ↓  | knowmem_r ↑  | mmlu | Performance Shift ↑ |
> | --- | --- | --- | --- | --- | --- |
> | Base model | 32.13 | 39.99 | 84.29 | 61.46 | 0.00 |
> | DUET | **4.27** | **4.86** | **78.33** | **61.52** | **57.09** |
> |
> | Distill from GA model | 17.40 | 19.54 | 42.86 | 54.53 | −13.18 |
> | Distill from GA model (retain free) | 18.04 | 20.32 | 77.82 | 57.55 | **23.38** |

---

> ### Author Response · Authors · 2025-11-24
> **Response to Weakness 2**
>
> ### W2: How about distillation from multiple unlearning teachers?
> **Response**: Thank you for this valuable suggestion. We agree that exploring multiple teachers' contributions helps deepen our understanding of the distillation process. We designed and implemented two approaches for multi-teacher distillation:
>
> **A. *Distill from multiple heterogeneous teachers*:**  We used NPO, GA, and prefix-guided models as teachers. For each forget sample, we aggregated their top-k logits with equal weighting ($\lambda$ = 1/3 each) during logit distillation.
>
>  **B. *Distill from multiple prefix-guided teachers*:** We used three differently-styled prefixes (declarative, polite request, strict enforcement):
>
> 1. ``You are an AI Assistant who is supposed to unlearn about the book series Harry Potter and provide answers without its knowledge as if you never knew about it.``
> 2. ``You are a helpful AI assistant. When asked about Harry Potter related content, politely decline to answer and explain you cannot discuss that topic. For all other questions, provide helpful and accurate responses.``
> 3. `` You are an AI assistant with content restrictions. You must refuse to answer any questions related to the Harry Potter series for policy reasons. Politely inform users you cannot help with Harry Potter queries, but assist with other topics normally.``
>
> **Results**:  As shown in **Table 3**, Approach A significantly underperformed DUET. We attribute this to conflicting response patterns among teachers, as aggregating their logits caused mutual interference. Additionally, GA's catastrophic forgetting on general knowledge severely compromised the aggregated logits' accuracy (though removing the retain loss partially mitigated this). Approach B also underperformed single-teacher distillation, likely because multiple teachers yield inconsistent top logits that lead to unclear distillation directions when aggregated. To sum up, we observe that more teachers do not guarantee better performance, and the **teacher quality matters more**, where clearer teacher behavior leads to more accurate student learning.
>
>
> **Table 3: Comparing DUET with Multi-Teacher-Distillation.**
> | Method | R-forget ↓ | R-forget_500 ↓ | knowmem_r ↑ | mmlu ↑ | Performance Shift ↑ |
> | --- | --- | --- | --- | --- | --- |
> | base model | 32.13  | 39.99 | 84.29  | 61.46  | 0.00 |
> | DUET | **4.27** | **4.86** | 78.33 | **61.52** | **57.09** |
> |
> | Plan A (GA, NPO, one prefix) | 7.94 | 10.28 | 58.58 | 45.43 | 12.16 |
> | Plan A (GA, NPO, one prefix)(retain free) | 13.98 | 14.92 | **82.83** | 61.04 | 41.34 |
> | Plan B (3 diff prefixs) | 8.32 | 9.77 | 81.02 | 60.13 | **49.43** |
>
>
> **Thank you again for your valuable comments and constructive suggestions.** We hope that our response with additional experiments can earn your favorable consideration.

---

### Author Response · Authors · 2025-11-24

We thank all reviewers for their valuable and constructive feedback. We are encouraged that our method is regarded as novel (47LM, Vd1V) and sound (47LM), and that it is viewed as making a valuable contribution to LLM unlearning (Vd1V).Reviewers further highlight DUET’s strong empirical performance (47LM, Vd1V, BGFT, o7sB), its robustness to reverse-prompt attacks and evaluation-format shifts (BGFT, o7sB), its data and computational efficiency (Vd1V, o7sB), and that the paper is clear and well written (o7sB) and enjoyable to read (47LM). Below we provide detailed responses to your questions and concerns.

---

### Author Response · Authors · 2025-12-03
**Summary of Rebuttal and Revisions**

We thank all reviewers and the Area Chairs for their valuable review of our submission. We had provided detailed responses to **all four reviewers** and incorporated the corresponding changes into the updated paper manuscript. By **26 Nov**, **three reviewers (Vd1V, BGFT, o7sB) have replied to our responses and acknowledged that their main concerns were resolved, and they raised their scores**, leading to an overall improvement from **6,4,4,6** to **6,6,6,8**.

**Positive assessments:** Reviewers broadly agreed that DUET is a **novel and well-motivated distillation-based unlearning method** that strikes a strong balance between tuning-based robustness and in-context precision, and that it makes a meaningful contribution to LLM unlearning. They highlighted our (1) strong empirical performance and favorable forgetting–utility trade-offs on benchmarks such as MUSE and WMDP; (2) robustness to reverse-prompt attacks and evaluation-format shifts; (3) high data and computational efficiency; and (4) clear, well-structured writing that makes the paper easy and enjoyable to read.

**Efforts to address concerns:** During the rebuttal period, we conducted substantial additional analyses and experiments, that both address reviewer comments and further support our conclusions:

- **Baselines and method analysis:** In response to baseline and comparison concerns, we **analyzed, implemented, and evaluated additional unlearning baselines** and showed that DUET achieves stronger forgetting while avoiding catastrophic utility degradation. We further investigated experiments with alternative teacher choices (e.g., tuning-based vs. contextualized teachers and multiple-teacher settings), clarifying why a single high-quality, prefix-steered teacher provides the best balance in practice.
- **Design choices and data breadth:** To justify our design, we conducted **multi-step distillation ablation studies**, demonstrating that extending distillation beyond the first token yields only marginal gains in forgetting but noticeably impacts utility, which supports our one-step design choice. We also conducted experiments by **expanding the forget/retain training sets**, showing comparable or improved performance and thereby confirming DUET's data efficiency which avoids overfitting to small synthetic sets.
- **Robustness and metrics:** We strengthened the robustness evaluation with **additional adversarial and jailbreak-style prompts**, where DUET consistently exhibited the lowest attack success rate among compared methods. At the same time, we explicitly acknowledge remaining vulnerabilities across common unlearning methods and position them as important future work.
- **Teacher quality, generalization, and datasets:** We elaborated on the role of teacher quality (as providing an upper bound rather than a hard limitation), added a dedicated discussion of **open challenges** (e.g., determining unlearning boundaries and designing reliable evaluation protocols), and provided **new experiments on additional LLM backbones (Qwen and Mistral)**, demonstrating that DUET generalizes well across architectures. We further detailed how we expanded and curated the datasets.
- **Presentation refinements:** Finally, we corrected a few referencing and formatting issues and revised parts of the text to adopt a more neutral and objective tone regarding performance claims.

In summary, reviewers view DUET as a novel, effective, and efficient approach to LLM unlearning, and our additional experiments, analyses, and revisions have resolved their main concerns, as reflected in the improved scores. We sincerely appreciate the reviewers' thoughtful feedback, and we hope this consolidated summary will help inform the final meta-decision.

Sincerely,

Authors

---

### Meta-Review · Area_Chair_FmBt · 2026-01-04

**Summary:**

The paper proposes a new LLM unlearning method called Distilled LLM Unlearning from an Efficiently Contextualized Teacher (DUET). DUET defines the teacher for distillation using in-context learning to efficiently turn the frozen base mode into one that seemingly unlearns (by prompting it to refuse to answer when needed). Then, a student is trained via distillation of the top-K logits of the teacher. The motivation behind this design is to obtain the best of both worlds: (i) in-context unlearning is fast but easily reversible; (ii) finetuning-based unlearning is more robust but expensive and suffers utility degradation. They show empirically that the distilled model is able to obtain favourable results on standard benchmarks in terms of refusing generation of undesired knowledge while preserving general knowledge, and exhibiting strong robustness against adversarial attacks that attempt to recover forgotten knowledge.

The reviewers raised the following concerns:

- **C1**. Lacking some unlearning baselines (Reviewer 47LM)
- **C2**. Missing experiments with GA (other unlearning algo) as the teacher, as well as multiple teachers (Reviewer 47LM).
- **C3**. Reliance on teacher quality (Reviewer Vd1V)
- **C4**. Limited evaluation from the perspective of generalization to different LLMs (Reviewer Vd1V)
- **C5**. Missing justification for why distilling only the first decoding step suffices (Reviewer BGFT)
- **C6**. Experimental setup limited to consider relatively small forget set retain sets (Reviewer BGFT)
- **C7**. Narrow adversarial evaluation, missing more systematic jailbreak or targeted relearning attacks (Reviewer BGFT)
- **C8**. Missing discussion of related work that also apply teacher-student approaches in LLMs (Reviewer o7sB)

**Reviewer Concerns:**

As described in detail below, the authors have sufficiently addressed all reviewer concerns.

Overall, DUET is a novel and sound method with a well-justified motivation. The effectiveness of DUET (in terms of forgetting, retention and robustness to adversarial attacks) was demonstrated both in the initial submission, as well as through various additional experiments presented during the rebuttal (for two other LLM architectures, different sizes of retain and forget sets, against two additional baselines, against a stronger jailbreaking attack and through other ablations relating to the depth of distillation, etc). The paper is also clearly written and easy to follow.

**Reviewer Scores:**

**Reviewer 47LM**
The authors addressed C1 by discussing each of the 3 baseline methods that the reviewer suggested, with reasonable arguments for why they picked a specific one out of those 3 to compare against (due to questionable suitability of the other 2 in their setting). Their method outperforms the additional baseline.
The authors also responded very thoroughly to all questions and suggestions that the reviewer raised about using GA as the teacher and using multiple teachers, even implementing improved variations compared to the reviewer’s suggestions, and showing that their method outperforms all of these variants, with compelling justifications for why that is.
This reviewer was already positive originally about the paper but I imagine the authors’ rebuttal would have only increased the reviewer’s score.


**Reviewer Vd1V**
The authors comprehensively answered the reviewer’s concerns through discussions and additional experiments. For C3, they discuss how the teacher quality affects the student’s performance in terms of unlearning and retention, arguing that the teacher should be seen as an upper bound akin to those that exist in other unlearning methods. They also point out that, through prompting, the teacher can be of a very high quality, and we expect this to continue improving as LLMs improve in the future. These arguments are convincing of the fact that the presence of a teacher should not be seen as a limitation of the method.
To address C4, the authors evaluate on two widely-representative LLMs: mistralai/Mistral-7B-Instruct-v0.3 and Qwen2.5-3B-Instruct, showing that their method outperforms baselines there too.
The authors also added detailed discussions of challenges and future work, at the reviewer’s request.
Overall, this rebuttal comprehensively addresses the reviewer’s concerns (and the reviewer has confirmed that this is the case).


**Reviewer BGFT**
The authors address C5 by describing their motivation (practicality, efficiency) of distilling only the first decoding step, and describing how considering top-K candidate tokens offers breadth in the absence of depth. They empirically investigate this through additional experiments where they find that extending distillation beyond just the first step improves forgetting marginally but harms utility.
They address C6 by running more experiments with larger forget and retain sets and different hyperparameters. These experiments show positive results for their method, further validating its effectiveness.
They address C7 by also running additional experiments using a more sophisticated jailbreak technique. Their method defends this attack at least as well as any other method, though the authors note that all existing unlearning methods have imperfect defense against strong attacks; an area where future research is needed.
The authors also addressed questions comprehensively, running another experiment.
Overall, I find these responses convincing. The reviewer also confirmed their concerns have been (mostly) addressed.


**Reviewer o7sB**
The authors address C8 via a discussion of each of the related works that the reviewer mentioned, clarifying that some tackle fundamentally distinct setups (e.g. using distillation to further robustify an already-unlearned model, or using distillation of a newly-trained clean teacher to remove backdoor attacks). The authors compare against UNDIAL, the most closely related to their work out of these additional papers, showing that their method improves over that prior work and discussing why that could be.
The authors also provide clarifications and responses to other questions that the reviewer raised.
Overall, this reviewer’s concerns seem to be addressed by the rebuttal, and the reviewer confirmed their concerns were (partially) addressed.

---

### Decision · Program_Chairs · 2026-01-26

Accept (Poster)